communications

biology

# Voltage-sensing phosphatase (Vsp) regulates endocytosis-dependent nutrient absorption in chordate enterocytes

Adisorn Ratanayotha [1,2,3], Makoto Matsuda[1], Yukiko Kimura[4], Fumiko Takenaga[1], Tomoaki Mizuno[5], Md. Israil Hossain[1], Shin-ichi Higashijima[4], Takafumi Kawai [1✉], Michio Ogasawara[6] & Yasushi Okamura [1,7✉]

Voltage-sensing phosphatase (Vsp) is a unique membrane protein that translates membrane electrical activities into the changes of phosphoinositide profiles. Vsp orthologs from various species have been intensively investigated toward their biophysical properties, primarily using a heterologous expression system. In contrast, the physiological role of Vsp in native tissues remains largely unknown. Here we report that zebrafish Vsp (Dr-Vsp), encoded by *tpte* gene, is functionally expressed on the endomembranes of lysosome-rich enterocytes (LREs) that mediate dietary protein absorption via endocytosis in the zebrafish mid-intestine. Dr-Vsp-deficient LREs were remarkably defective in forming endosomal vacuoles after initial uptake of dextran and mCherry. Dr-Vsp-deficient zebrafish exhibited growth restriction and higher mortality during the critical period when zebrafish larvae rely primarily on exogenous feeding via intestinal absorption. Furthermore, our comparative study on marine invertebrate *Ciona intestinalis* Vsp (Ci-Vsp) revealed co-expression with endocytosis-associated genes in absorptive epithelial cells of the *Ciona* digestive tract, corresponding to zebrafish LREs. These findings signify a crucial role of Vsp in regulating endocytosis-dependent nutrient absorption in specialized enterocytes across animal species.

[1] Laboratory of Integrative Physiology, Department of Physiology, Graduate School of Medicine, Osaka University, Suita, Osaka 565-0871, Japan. [2] Institute for Transdisciplinary Graduate Degree Programs, Osaka University, Suita, Osaka 565-0871, Japan. [3] Department of Anatomy, Faculty of Medicine Siriraj Hospital, Mahidol University, Bangkoknoi, Bangkok 10700, Thailand. [4] Exploratory Research Center on Life and Living Systems and National Institute for Basic Biology, National Institutes of Natural Sciences, Okazaki, Aichi 444-8787, Japan. [5] Center for Medical Research and Education, Osaka University, Suita, Osaka 565-0871, Japan. [6] Department of Biology, Graduate School of Science, Chiba University, Inage-ku, Chiba 263-8522, Japan. [7] Graduate School of Frontier Biosciences, Osaka University, Suita, Osaka 565-0871, Japan. ✉email: kawai@phys2.med.osaka-u.ac.jp; vsop1@me.com

Phosphoinositides (PIPs) are essential phospholipids that constitute eukaryotic biological membranes. Differential phosphorylation of inositol head groups at the 3-, 4-, and 5-phosphate positions results in seven PIP species: three mono-phosphorylated PIPs, three bi-phosphorylated PIPs, and a single tri-phosphorylated PIP. These interconvertible PIP species are distinctly localized at the plasma membrane and subcellular compartments, regulating specific membrane-associated activities within the cells, including membrane dynamics and cellular trafficking[1–5]. Because of their essential roles, PIPs have attracted research attention for their regulatory mechanism and fundamental interplay in cell physiology.

Voltage-sensing phosphatase (Vsp, also known as Tpte) is among the key molecules that regulate PIPs' homeostasis. Encoded by *tpte* gene, Vsp is a unique membrane protein with two functional domains: the voltage sensor domain (VSD), as typically found in voltage-gated ion channels; and the cytoplasmic catalytic region sharing molecular similarity to the phosphatase and tensin homolog deleted on chromosome 10 (PTEN), a tumor-suppressing PIP phosphatase. Vsp functions upon membrane depolarization to exhibit voltage-dependent phosphatase activity towards PIPs; thus, directly translating membrane electrical activities into intracellular PIP signals. Unlike PTEN, which exhibits rigid 3-phosphatase activity toward $PI(3,4,5)P_3$ and $PI(3,4)P_2$, Vsp exhibits activities of both 3- and 5-phosphatases, thereby mediating four subreactions: dephosphorylating $PI(3,4,5)P_3$ to $PI(4,5)P_2$ or $PI(3,4)P_2$; and from $PI(4,5)P_2$ or $PI(3,4)P_2$ to $PI(4)P$[6–8]. Among four subreactions, the reaction with $PI(4,5)P_2$ to produce $PI(4)P$ is the most robust[7,9]. Vsp orthologs are widely conserved across animal species, and their biophysical properties have been intensively investigated using in vitro approach. Vsp expression has been detected in various animal tissues, including testis[10–12], epithelial cells of digestive tract and renal tubules[13–16], neurons[6,17], and blood cells[13]. Our recent study has identified that mouse Vsp is required for normal sperm motility through its role in regulating the spatial distribution of $PI(4,5)P_2$ in sperm flagellum[11]. However, the physiological function of Vsp in most organs remains elusive.

Previous studies demonstrated that zebrafish *Danio rerio* Vsp (Dr-Vsp or Dr-Tpte, hereafter referred to as Dr-Vsp) shares similar molecular architecture to other known Vsp proteins and preserves their key biophysical properties[18,19]. Zebrafish show conserved physiology and anatomy with mammals[20]. Its optical transparency during early development enables non-invasive monitoring and in vivo analysis of cellular behaviors[21,22]. By taking these advantages of the zebrafish, we studied the physiological functions of Vsp at the whole animal level. We show that Vsp is highly expressed in lysosome-rich enterocytes (LREs), which are specialized enterocytes that contain large supranuclear vacuoles and absorb dietary protein from the intestinal lumen[23–26]. We present evidence from in vivo assays that Vsp contributes to the endocytosis of enterocytes. Comparative analysis on ascidian *Ciona intestinalis* Type A also reveals co-expression of Vsp (Ci-Vsp) with the endocytic adapter Dab2 (Ci-Dab2) in absorptive epithelial cells of the digestive tract, suggesting that the function of Vsp in enterocytes is evolutionarily conserved. These results highlight a crucial role of Vsp in the endocytosis-dependent nutrient absorption of intestinal epithelial cells.

## Results

**Dr-Vsp is expressed in the zebrafish intestine**. RT-PCR analysis revealed substantial expression of Dr-Vsp mRNA in the testis, spleen, intestine, kidney, and gills of adult zebrafish (Fig. 1a). Dr-Vsp mRNA was also detected in 7-dpf zebrafish larvae (Fig. 1b).

Whole-mount in situ hybridization (WISH) showed specific expression of Dr-Vsp mRNA in the mid-intestine (Fig. 1c), with the signal detectable as early as 5 days after fertilization (dpf). The result was comparable to the EGFP signal observed in our *Tg(tpte:EGFP)* transgenic zebrafish (Fig. 1d, e), which were generated using CRISPR-Cas9-mediated knock-in transgenesis[27] to express EGFP recapitulating endogenous expression of *tpte*, the Dr-Vsp-encoding gene (Supplementary Fig. 1). These findings suggest a potential role of Dr-Vsp in the digestive tract from an early developmental stage.

**Dr-Vsp is spatially distributed inside zebrafish enterocytes**. Dr-Vsp protein expression was examined using an anti-Dr-Vsp antibody (NeuroMab clone N432/21). The immunostainings detected abundant expression of Dr-Vsp protein in the epithelial cells of larval pronephros and the posterior portion of the mid-intestine (Fig. 2a), correlating with the results of WISH and *Tg(tpte:EGFP)* transgenesis.

Here we focused on Dr-Vsp expression in the intestine. Principal enterocytes in this region have been characterized as lysosome-rich enterocytes (LREs) that mediate dietary protein absorption via endocytosis[23–26]. Dr-Vsp immunostaining in the intestine of *Tg(tpte:EGFP)* transgenic zebrafish revealed complete co-localization of Dr-Vsp signal and EGFP-positive LREs (Fig. 2b). Notably, Dr-Vsp expression is absent in the intestinal epithelial cells (IECs), which morphologically differ from LREs by the absence of large supranuclear vacuoles (Fig. 2c, d)[28]. We also analyzed the transcriptome database of zebrafish LREs published by Park et al.[23] and found that Dr-Vsp is highly enriched in LREs (Fig. 2e), along with other genes known to be associated with macromolecule absorption, such as *lamp2*, *cubn*, and *dab2*[23,29].

The positive signal of the anti-Dr-Vsp antibody was not entirely colocalized with phalloidin staining, which represents actin filaments (F-actin) in microvilli (Fig. 3a, b), indicating that Dr-Vsp in LREs is spatially distributed at intracellular region beneath the apical surface rather than at the cell surface or microvilli. Quantification of fluorescence (Fig. 3c) demonstrates that Dr-Vsp showed the most intense signal in a deeper region of the cells than the microvillous F-actins. Immuno-electron microscopy analysis revealed a particularly high density of immunoparticles at the subapical region of LREs, while it is rarely detectable at microvilli (Fig. 4a and Supplementary Fig. 2), corresponding to the immunofluorescence results. To specify which intracellular structures contain Dr-Vsp, we used the MDCKII cells as a model for examining the intracellular compartments involved in epithelial endocytosis[30], simulating zebrafish LRE function. In MDCKII cells, Dr-Vsp was co-expressed with (i) Rab5, an early endosome marker; or (ii) Rab11, a recycling endosome marker. The Dr-Vsp signal was broadly colocalized with Rab5-positive and Rab11-positive intracellular compartments (Fig. 4b and Supplementary Movies 1, 2), suggesting that Dr-Vsp is potentially expressed on the membranes of early and recycling endosomes.

**Dr-Vsp contributes to the survival and growth of zebrafish larvae**. To investigate Dr-Vsp function under physiological conditions, we generated Dr-Vsp-deficient (Dr-Vsp$^{-/-}$) zebrafish using CRISPR-Cas9-mediated mutagenesis, targeting the early exon of the *tpte* gene. Mutant zebrafish lacking the transmembrane segments and cytoplasmic catalytic region of Dr-Vsp were obtained (Supplementary Fig. 3).

Dr-Vsp$^{-/-}$ larvae showed normal gross morphology and behavior. However, we noticed that the survival rate of Dr-Vsp$^{-/-}$ mutants was significantly lower than that of wild-type larvae at the age of 10–14 dpf (Fig. 5a). During this period, the yolk is completely

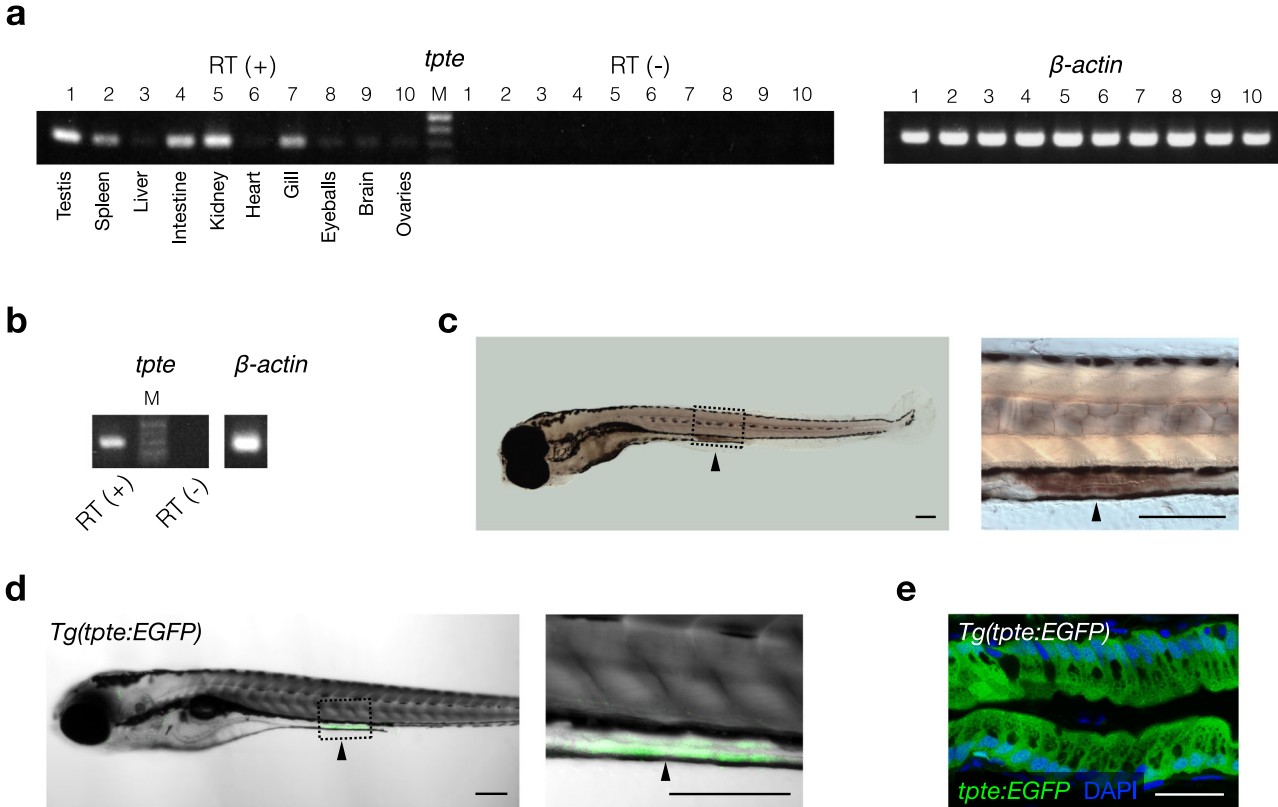

**Fig. 1 Expression profile of the Dr-Vsp-encoding gene (*tpte*) in zebrafish tissues. a** RT (+), *tpte* expression in adult zebrafish tissues: (1) testis, (2) spleen, (3) liver, (4) intestine, (5) kidney, (6) heart, (7) gill, (8) eyeballs, (9) brain, and (10) ovaries. RT (−), negative control for all tissues in the same order. *β-actin*, positive control. M, marker. **b** RT (+), *tpte* expression in 7-dpf zebrafish larva. RT (−), negative control. *β-actin*, positive control. M, marker. **c** Whole-mount in situ hybridization (WISH) of 6-dpf zebrafish larva showing *tpte*-positive signal (arrowhead) in enterocytes of mid-intestine. Scale bar = 200 μm. **d** A 5-dpf *Tg(tpte:EGFP)* zebrafish larva expressing EGFP under the promotor of the Dr-Vsp-encoding gene in enterocytes of mid-intestine (arrowhead). The EGFP-positive cells in this figure correspond to the *tpte*-positive cells in **c**. Green, Dr-Vsp (*tpte:EGFP*). Scale bar = 200 μm. **e** Confocal image of larval enterocytes at mid-intestine of a 5-dpf *Tg(tpte:EGFP)* zebrafish larva (sagittal section). The enterocytes in this region are known as lysosome-rich enterocytes (LREs) that contain large supranuclear vacuoles. Green, Dr-Vsp (*tpte:EGFP*). Blue, DAPI. Scale bar = 25 μm.

reabsorbed, and zebrafish growth is generally dependent on exogenous feeding via the digestive system[28,31]. Moreover, Dr-Vsp[−/−] mutants that survived from the early period exhibited long-term growth restriction, as evidenced by their smaller size compared to their sibling wild types (Fig. 5b, c). Despite the absence of obvious congenital deformities, these results could be inferred that Dr-Vsp function contributes to the survival and growth of zebrafish larvae during early life. The remaining Dr-Vsp[−/−] mutants thrived to adulthood and reached reproductive maturity at around the same age as wild-type zebrafish without apparent fertility defects.

**Dr-Vsp facilitates endocytic nutrient absorption in larval LREs.** Dr-Vsp is abundant at the endomembranes inside LREs, which absorb nutrients via endocytosis[23–26], suggesting that Dr-Vsp could be involved in this specialized process. To test this hypothesis, we compared the endocytosis efficiency between the wild type and Dr-Vsp[−/−] larvae by gavaging 6-dpf zebrafish larvae with either fluorescent dextran (fDex) or mCherry solution directly into the digestive tract (Fig. 6a)[32] and subsequently observed endocytosis in LREs using confocal microscopy. fDex was used for assessing fluid-phase endocytosis; and mCherry was used as a protein cargo for assessing receptor-mediated endocytosis[23,29].

At 2 h after gavage, fDex and mCherry were absorbed into LREs in both wild-type and Dr-Vsp[−/−] zebrafish larvae (Fig. 6b, c). However, the uptake of fDex and mCherry were reduced in Dr-Vsp[−/−] LREs (Fig. 6d, e). In the experiment with fDex, the

fluorescent signals in Dr-Vsp[−/−] LREs were mostly confined to apical vesicles, with less signal found in larger endosomes and supranuclear vacuoles (Fig. 6d). The average percentage of LREs containing fDex-filled endosomal vacuoles was nearly 80% in wild types but less than 30% in Dr-Vsp[−/−] larvae (Fig. 6f), reflecting that Dr-Vsp[−/−] mutants have a lower proportion of LREs capable of internalizing fDex and forming subsequent endosomal vacuoles. We further analyzed the relative fDex internalization profile of LREs by measuring fluorescent intensity within the cells (Fig. 6g)[23,29]. As shown in Fig. 6g, fDex uptake into Dr-Vsp[−/−] LREs was clearly impaired both in the apicobasal axis of individual cells and the anteroposterior axis across longitudinal segments of LREs. Similar results were obtained in the mCherry experiment (Fig. 6e): the average percentage of LREs containing mCherry-filled endosomal vacuoles was greater than 80% in wild types versus 40% in Dr-Vsp[−/−] larvae (Fig. 6h). Spatial profile of the marker distribution in the cells (Fig. 6i) also showed reduced uptake of mCherry in Dr-Vsp[−/−] LREs.

When we gavaged wild-type zebrafish with the mixture of fDex and mCherry, we found that while most LREs contained both fluorescent cargoes in the same supranuclear vacuoles, some vacuoles contained only fDex or mCherry (Supplementary Fig. 4a). In Dr-Vsp[−/−] mutants, on the other hand, mCherry was transported into some supranuclear vacuoles, whereas fDex was mostly found in apical vesicles inside Dr-Vsp[−/−] LREs (Supplementary Fig. 4b). This suggests that the uptake defect in Dr-Vsp[−/−] LREs was more severe with fDex than with mCherry.

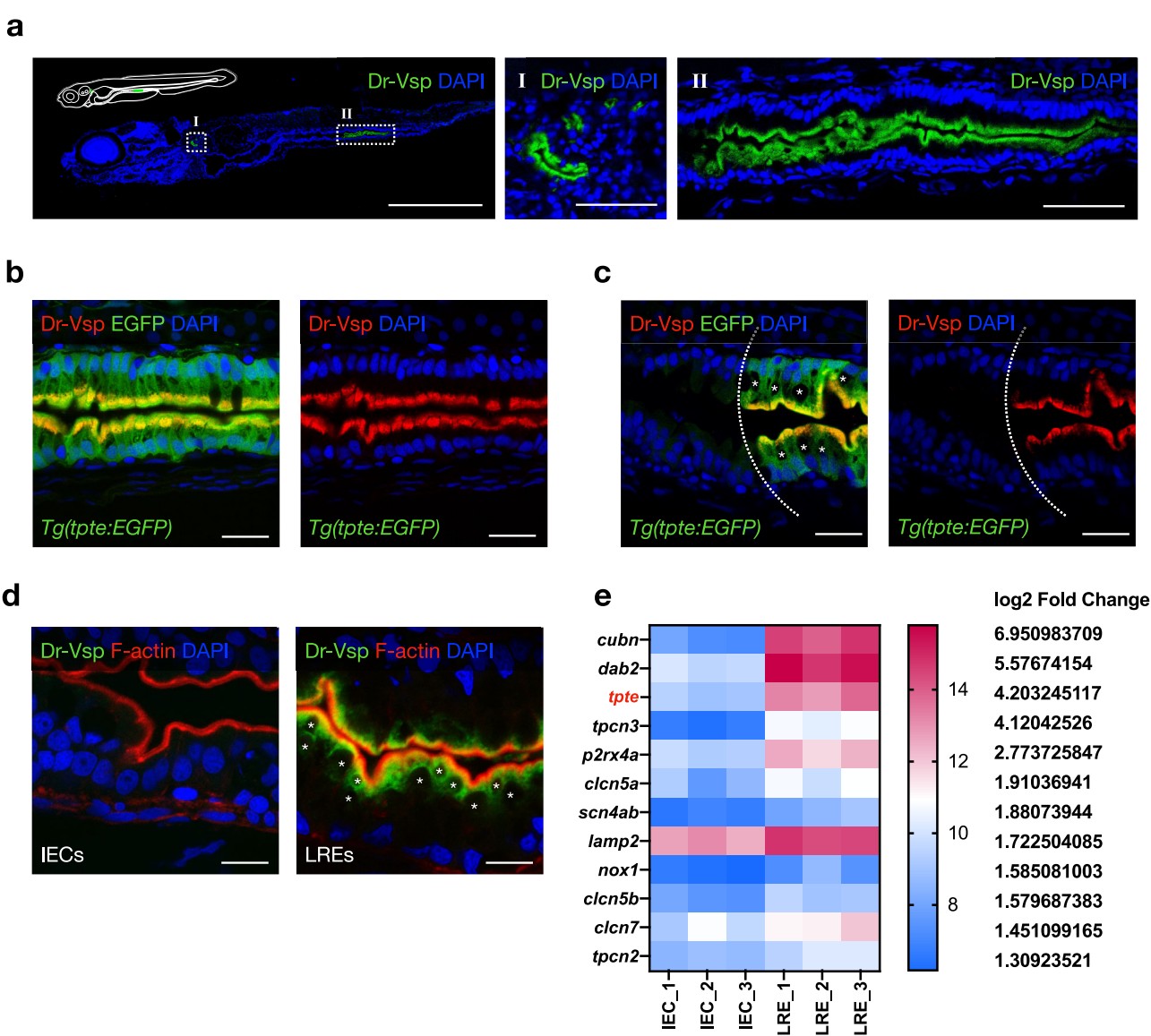

**Fig. 2 Expression profile of Dr-Vsp protein in zebrafish tissues. a** Confocal images of Dr-Vsp immunostaining in a 7-dpf wild-type zebrafish larva (sagittal section). Dr-Vsp is highly expressed in (I) larval pronephros and (II) enterocytes at the posterior part of the mid-intestine, which are defined as lysosome-rich enterocytes (LREs). Green, Dr-Vsp. Blue, DAPI. Scale bar = 500 μm (left) and 50 μm (middle and right). **b** Dr-Vsp immunostaining in LREs of a 7-dpf *Tg(tpte:EGFP)* transgenic zebrafish larva (sagittal section). Dr-Vsp expression signals were completely colocalized with EGFP signals in LREs. Green, Dr-Vsp *(tpte:EGFP)*. Red, Dr-Vsp (antibody). Blue, DAPI. Scale bar = 20 μm. **c** Dr-Vsp immunostaining in a 7-dpf *Tg(tpte:EGFP)* transgenic zebrafish larva (sagittal section), showing a border between intestinal epithelial cells (IECs; Vsp-negative) and LREs (Vsp-positive). LREs contain large supranuclear vacuoles in the cytoplasm, contrasting with IECs. Green, Dr-Vsp *(tpte:EGFP)*. Red, Dr-Vsp (antibody). Blue, DAPI. Scale bar = 20 μm. The background signal of the EGFP channel was increased in the left figure to enhance the visibility of IECs. **d** Dr-Vsp immunostaining in IECs (left) and LREs (right) of a 7-dpf wild-type zebrafish larva (sagittal section). Dr-Vsp expression was restricted mainly to LREs, but not detectable in other IECs. Green, Dr-Vsp. Red, F-actin. Blue, DAPI. Scale bar = 10 μm. Images were visualized under an LSM880 confocal microscope with AiryScan. **e** Heatmap of RNA-seq data representing expression levels of selected genes in IECs and LREs. Raw data of RNA-seq experiments were retrieved from the GEO Datasets database with accession number GSE124970 contributed by ref. [23] and analyzed using the web-based application iDEP v0.92[69].

Taken together, these findings support our hypothesis that Dr-Vsp is involved in the endocytosis of LREs, specifically in the maturation of endosomal vacuoles following initial endocytosis. The findings also raise the possibility that Dr-Vsp potentially plays a greater role in fluid-phase endocytosis, the primary mechanism of fDex uptake into cells.

**Dr-Vsp maintains intact LRE morphology.** We examined the LRE morphology of 14-dpf zebrafish larvae, the age around 1 week after zebrafish enterocytes completed morphogenesis (Fig. 7a)[28,31]. This age also corresponds to the period when higher mortality was observed in Dr-Vsp$^{-/-}$ larvae (Fig. 5a). We measured key parameters that represent cell morphology: central height, apical width, basal width, and microvillus length[33]. The measurement revealed that Dr-Vsp$^{-/-}$ LREs were significantly shorter in all other parameters except for the microvillus length (Fig. 7b). The fact that Dr-Vsp deficiency primarily affected the cell body is consistent with the idea that Dr-Vsp plays a major role at the endomembranes within cells than at the plasma membrane or microvilli. The defect in LRE morphology was not observed at early stages, such as at 6 dpf (Fig. 7c). We examined histological sections of 14-dpf zebrafish LREs with toluidine blue

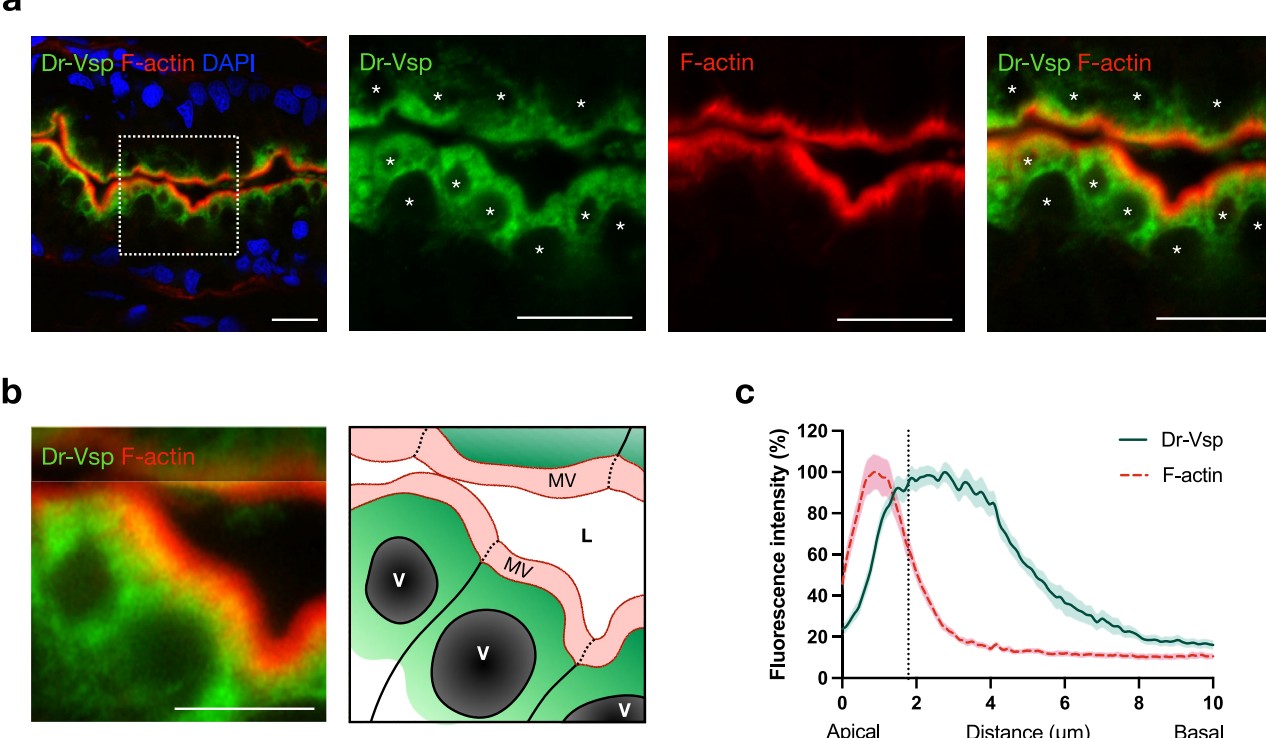

**Fig. 3 Expression profile of Dr-Vsp protein at the subapical region of LREs. a** Confocal images of Dr-Vsp immunostaining showing the subcellular distribution of Dr-Vsp in larval LREs. Dr-Vsp is primarily localized in the cytoplasm underneath the apical surface but barely detectable at the microvilli and plasma membrane of LREs. Green, Dr-Vsp. Red, F-actin, Blue, DAPI. *Vacuole. Scale bar = 10 μm. **b** Confocal image of larval LREs at higher magnification and their corresponding schematic illustrations, focusing on the intracellular distribution of Dr-Vsp. Green, Dr-Vsp. Red, F-actin. V vacuole, MV microvilli, L lumen. Scale bar = 5 μm. **c** Fluorescence intensity of Dr-Vsp (green) and F-actin (red) along the apicobasal axis of individual LREs. Data were presented as mean ± SEM. Enterocytes = 16 cells from 4 larvae for each zebrafish line. The dotted line in the graph conceptually indicates the junction between the microvilli and the cell body.

staining and analyzed endosomal vacuoles, comparing wild-type and Dr-Vsp$^{-/-}$ mutants (Fig. 7d). Although supranuclear vacuoles could be seen in most cells, the measurement of vacuolar perimeter and area showed that their size was apparently smaller in Dr-Vsp$^{-/-}$ LREs (Fig. 7e, f). We also used transmission electron microscopy (TEM) to define the ultrastructures of 14-dpf LREs, focusing on the subapical region where Dr-Vsp is typically expressed. TEM images of wild-type and Dr-Vsp$^{-/-}$ LREs (Fig. 7g) revealed key features of absorptive enterocytes, such as membrane invaginations at inter-microvillous spaces, cytoplasmic tubules, branching tubules, and endocytic vesicles associated with invaginated membranes (tubule-vacuole complexes)[24,34,35]. Interestingly, these features appeared less prominent in Dr-Vsp$^{-/-}$ LREs. Further observations suggested that small endocytic vesicles in Dr-Vsp$^{-/-}$ LREs are located in deeper regions than those in wild types, increasing the distance between the apical surface and vesicle-dense area (Supplementary Fig. 5). Moreover, branching tubules and tubule-vacuole complexes were less frequently present in the subapical region of mutant LREs (Supplementary Fig. 6), seemingly reflecting the disturbance in endosome maturation. Collectively, these results suggest that LREs require Dr-Vsp in regulating the formation and expansion of endosomal vacuoles, which in turn maintain intact LRE morphology. Figure 7h summarizes how Dr-Vsp contributes to endosomal vacuoles and the morphology of LREs.

**Endocytic role of Vsp is potentially conserved in the ascidian digestive tract.** Vsp gene is widely conserved among chordates (Supplementary Figs. 7, 8). To determine if the role of Vsp in the endocytosis of enterocytes is conserved among animal species, we investigated the expression profile of sea squirt *C. intestinalis* Vsp (Ci-Vsp) in the digestive tract. Sea squirt belongs to the basal chordates, and the gene expression profiles of *C. intestinalis* Type A were studied in great detail with reference to chordate evolution[36]. WISH on transparent 2-week-old *Ciona* juveniles revealed the expression of Ci-Vsp mRNA in the stomach, mid-intestine, and posterior-intestine (Fig. 8a), which were known as absorptive regions of the *Ciona* digestive tract[37]. Higher magnification showed a strip-like expression pattern both in the stomach and bulged intestine. Similar patterns were found in the digestive tract of *Ciona* young adults, with the expression signal being more robust in the stomach and mid-intestine than in the posterior-intestine (Fig. 8b). Additional Ci-Vsp mRNA expression was observed in the looped region of the intestine (Fig. 8b, white arrowhead), which corresponds to a region expressing several absorptive genes, including peptide transporter (*PEPT1*) and monosaccharide transporters (*SGLT1* and *GLUT5*) (Supplementary Fig. 9).

Transverse sections of the digestive tract obtained from *Ciona* juveniles (Fig. 8c) showed that the expression signals were present alternately in the digestive epithelial cells. In the stomach, Ci-Vsp signals were observed mainly in the inner-folds, where the epithelial cells predominantly express absorptive genes[38]. In the outer-folds of the stomach, known to express many pancreatic-related exocrine genes[39], Ci-Vsp signals were occasionally found in the regions which seemed to start invagination of newly generated inner-folds. Similarly, in the bulged-region of the mid-intestine, circumferential epithelial cells expressed Ci-Vsp in a pattern corresponding to the alternating folding morphology present in the stomach.

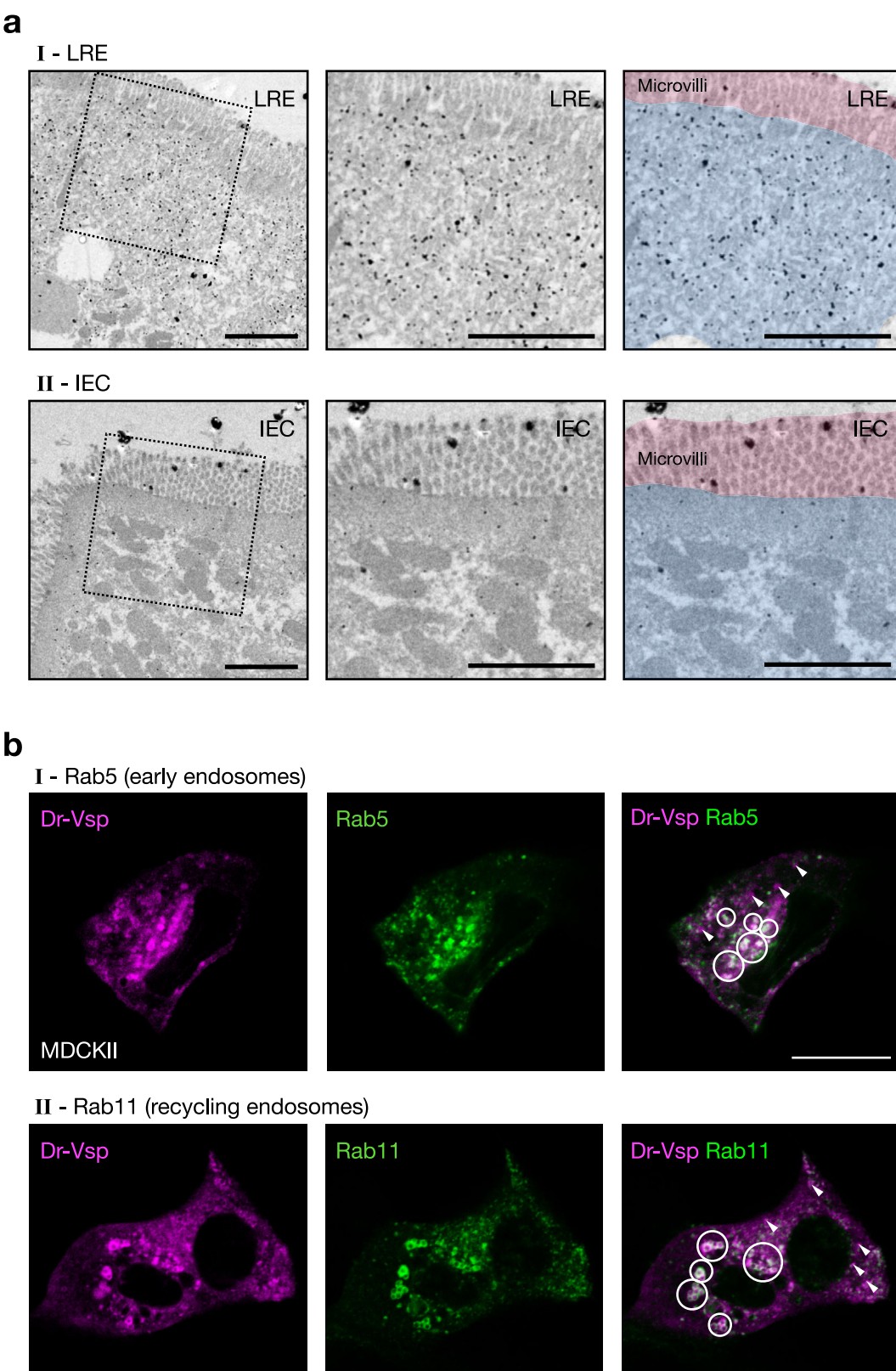

We further examined whether Ci-Vsp expression correlates with endocytosis-associated genes. Several genes involved in enteric endocytosis in the zebrafish model have previously been identified, including Cubilin (*cubn*), Amnionless (*amn*), and Dab2 (*dab2*)[23]. In *Ciona*, however, only *dab2* ortholog (Ci-Dab2) is expressed as a single homolog, so Ci-Dab2 was selected to represent the endocytosis-associated gene in this study. In WISH on 2-week-old *Ciona* juveniles, the Ci-Dab2 expression pattern is highly comparable to that of Ci-Vsp (Fig. 8d–f), implying their concerted function in the *Ciona* digestive tract. Ci-Vsp is co-

**Fig. 4 Expression profile of Dr-Vsp protein at endomembranes. a** Pre-embedding silver-enhanced immunogold staining for Dr-Vsp in (I) LREs and (II) IECs of 14-dpf wild-type zebrafish larva. The right two images are the same as the middle images but were highlighted with red and blue for microvilli and subapical region, respectively. In LREs, immunoparticles for Dr-Vsp are preferentially distributed at the subapical region while rarely detectable at microvilli, consistent with results of Dr-Vsp immunofluorescence in Fig. 3. In IECs, immunoparticles for Dr-Vsp are rarely detectable at both subapical region and microvilli, consistent with Dr-Vsp immunofluorescence and RNA-seq analysis in Fig. 2. Scale bar = 2 μm. **b** MDCKII cells co-expressing Dr-Vsp-mCherry and (I) Rab5-EGFP for early endosomes; or (II) Rab11-EGFP for recycling endosomes. Colocalized fluorescent signals (circles) represent Dr-Vsp expression at the endosomal membranes of early and recycling endosomes. Note that non-colocalized Dr-Vsp signals (arrowheads) are also present in both images. Magenta, Dr-Vsp. Green, Rab5 or Rab11. Scale bar = 20 μm.

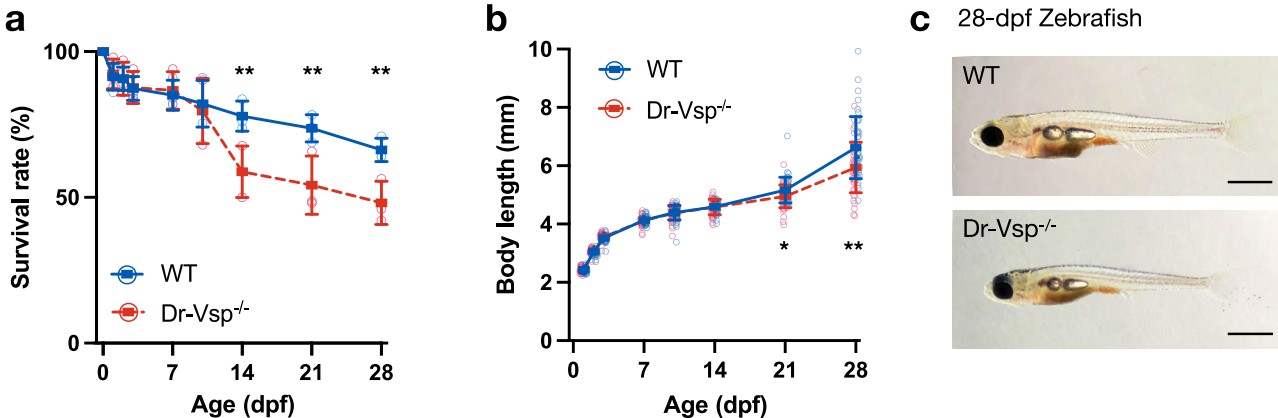

**Fig. 5 Phenotypic investigation of Dr-Vsp⁻/⁻ zebrafish. a** Survival rates of wild-type and Dr-Vsp⁻/⁻ zebrafish larvae during the first month of age. Zebrafish larvae, 300 at D0 for each zebrafish line. All statistical data were collected from three independent experiments. **b** Body length of wild-type and Dr-Vsp⁻/⁻ zebrafish larvae raised under a non-calorie-restricted standard diet. Zebrafish larvae on the first day (D0) is 45 for wild-type and 44 for Dr-Vsp⁻/⁻ zebrafish. Error bars, means ± SD; *$P < 0.05$; **$P < 0.01$; unpaired Student's $t$-test. **c** Brightfield images of the representative wild-type (WT) and Dr-Vsp⁻/⁻ zebrafish larvae at 28 dpf, raised under the same conditions. Scale bar = 1 mm.

expressed with Ci-Dab2 in a strip-like pattern in the stomach and bulged intestine, indicating that Ci-Vsp is expressed in the epithelial cells with endocytosis-dependent absorptive function.

## Discussion

Vsp expression was previously reported in the digestive organs of other animal species, such as in the gut epithelial cells of invertebrate *C. intestinalis* Type A[13] and the stomach of chicken embryos[17]. However, the functional role of Vsp in enterocytes has not been studied in detail. We report here the expression of Vsp in specialized enterocytes known as LREs in the mid-intestine of zebrafish. Vsp in LREs is most likely expressed on the endosomal membranes required for endocytosis. Loss of Vsp activity impaired LRE endocytosis and nutrient absorption, potentially accounting for growth restriction and increased mortality in developing zebrafish. Dr-Vsp was detectable in LREs as early as 5 dpf (Figs. 1, 2), coinciding with the time when zebrafish intestine begins nutrient absorption[40,41]. These findings highlight an essential physiological role of Vsp in endocytosis-dependent nutrient absorption besides its previously established role in sperm physiology[11].

Previous research indicated that Vsp is expressed on the plasma membrane, based on studies of native expression in *Ciona* sperm[6], *Xenopus* renal epithelial cells[14,16], and mouse sperm[11], as well as in vitro experiments using heterologous expression system[6,15,42,43]. The same was reported for Dr-Vsp when it was first characterized by whole-cell patch-clamp recordings of mammalian cells heterologously expressing Dr-Vsp protein[18]. However, our results of immunofluorescence experiments and immuno-electron microscopy revealed that native Dr-Vsp in LREs was localized to the subapical region but not on the microvilli nor plasma membrane (Figs. 3, 4a). Results from the

MDCK model seem to suggest that Dr-Vsp is enriched at the early and recycling endosomal membranes (Fig. 4b), consistent with our recent in vitro study in which fluorescent signal from mCherry-fused Dr-Vsp expressed in HEK293T cells was also distributed throughout the intracellular compartments[19]. Although additional studies, including in vivo co-localization analysis with endosome markers in LREs, are required, our current findings provide converging evidence that Dr-Vsp functions primarily on endomembranes at the subapical region of LREs.

PIP conversion has been well documented to regulate membrane trafficking in the endosomal system, and this process is achieved by PIP-metabolizing enzymes, including PIP kinases and phosphatases[3–5]. Dr-Vsp functions as 5-phosphatase and 3-phosphatase, allowing it to hydrolyze PI(3,4,5)P₃, PI(4,5)P₂, and PI(3,4) P₂[7,9]. Following gavage with fDex and mCherry, we observed that endocytosis was remarkably decreased in Dr-Vsp⁻/⁻ LREs, as evidenced by the reduction of subsequent endosomal vacuoles and fluorescence uptake into the cells (Fig. 6). Histological and TEM findings show that loss of Dr-Vsp resulted in fewer and smaller vacuoles and, consequently, a reduction in the overall cell volume of LREs. Besides, Dr-Vsp⁻/⁻ LREs show reduced numbers of endocytic vesicles, cytoplasmic tubules, and tubule-vacuole complexes in the subapical region (Fig. 7 and Supplementary Figs. 5, 6). These findings suggest that Dr-Vsp is involved in endocytic regulation, and Dr-Vsp deficiency disrupts membrane trafficking in LREs.

Dr-Vsp deficiency may alter PIP homeostasis on various types of endomembranes, which play a critical role in membrane trafficking. However, it remains a challenge to determine which enzyme subreaction mediated by Vsp contributes to the phenotypes in LREs. Here we propose potential scenarios based on Dr-Vsp's dual phosphatase activities. One scenario considers a more robust Dr-Vsp's 5-phosphatase activity toward PI(4,5)P₂ and PI(3,4,5)P₃[9]. We speculate that intracellular Dr-Vsp

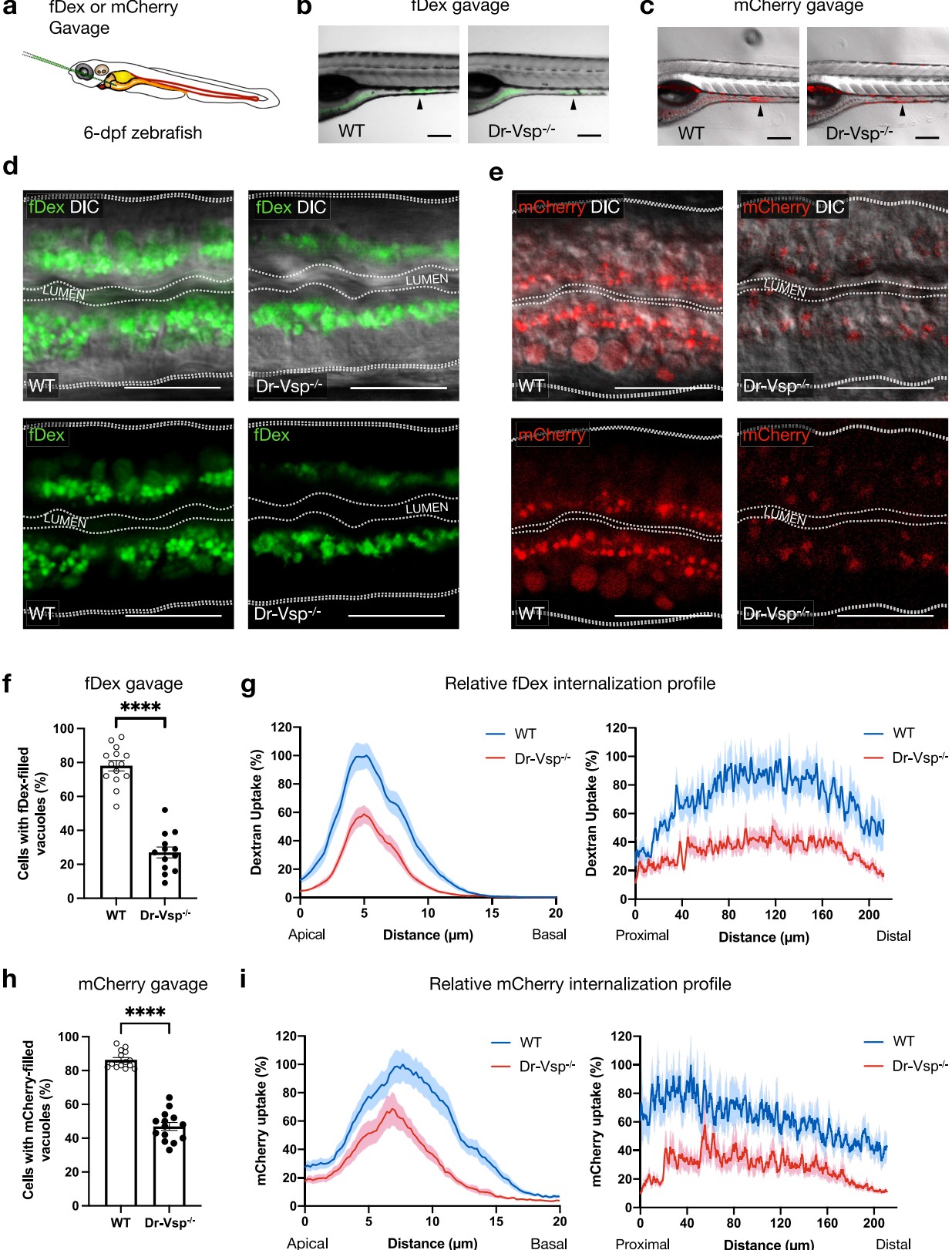

dephosphorylates such PIPs on the membranes of newly enclosed endocytic vesicles during the transformation to early endosomes[44–46]. Defective endocytosis in $Dr\text{-}Vsp^{-/-}$ LREs is likely caused by reduced dephosphorylation of endosomal PI(4,5) $P_2$ and possibly PI(3,4,5)$P_3$, leading to ectopic accumulation of these PIPs that prevent apical endocytic vesicles from transforming into, or fusing with, subsequent intracellular compartments during endosomal maturation; thus restricting most fDex and mCherry to the subapical region of $Dr\text{-}Vsp^{-/-}$ LREs. It should be noted that other 5-phosphatase, such as OCRL1 and INPP5B, can be detected in LREs, although at lower levels according to RNA-seq data[23]. These enzymes are involved in

**Fig. 6 Functional analysis of Dr-Vsp in larval enterocytes. a** Experimental approach: 6-dpf zebrafish larvae were gavaged with Alexa Fluor™ 488-tagged dextran (fDex) or mCherry solutions into the anterior intestinal bulb. Live imaging was performed at 2 h after gavage. **b–e** fDex and mCherry internalized into zebrafish LREs after gavage. **b** Live confocal images of wild-type (WT) and Dr-Vsp$^{-/-}$ zebrafish larvae showing fDex internalization in enterocytes at the posterior portion of mid-intestine (arrowhead). Scale bar = 200 μm. **c** Live confocal images of wild-type and Dr-Vsp$^{-/-}$ zebrafish larvae showing mCherry internalization in enterocytes at the posterior portion of mid-intestine (arrowhead). Scale bar = 200 μm. **d** Live confocal images of wild-type and Dr-Vsp$^{-/-}$ enterocytes showing fDex internalization. The images were presented with DIC (top) or without DIC (bottom) to enhance the visibility of the internalized fDex. Dotted lines conceptually indicate the outlines of larval intestines. Scale bar = 20 μm. **e** Live confocal images of wild-type and Dr-Vsp$^{-/-}$ enterocytes showing mCherry internalization. The images were presented with DIC (top) or without DIC (bottom) to enhance the visibility of the internalized mCherry. Dotted lines conceptually indicate the outlines of larval intestines. Scale bar = 20 μm. **f, g** Endocytosis efficiency analysis in LREs after fDex gavage. **f** Percentage of the enterocytes containing fDex-filled vacuoles at 2 h following gavage, representing the proportion of cells capable of internalizing fDex and forming subsequent endosomal vacuoles. Error bars, means ± SEM; ****$P < 0.0001$; unpaired Student's $t$-test. Number of enterocytes = 344 cells from 14 wild-type and 359 cells from 13 Dr-Vsp$^{-/-}$ zebrafish. **g** Relative fDex internalization profile comparing between wild-type and Dr-Vsp$^{-/-}$ zebrafish larvae at 2 h following gavage. Data were means ± SEM percentage of intracellular fDex intensity along the specified axes, with 100% indicating the maximal value of the average intensity on each axis. (Left) Apicobasal axis of individual cells. (Right) Longitudinal axis along the posterior portion of mid-intestine. Number of fish = 14 for wild-type and 13 for Dr-Vsp$^{-/-}$. Data were presented as mean ± SEM, collected from four independent experiments for both wild-type and Dr-Vsp$^{-/-}$. **h, i** Endocytosis efficiency analysis in LREs after mCherry gavage. **h** Percentage of the enterocytes containing mCherry-filled vacuoles at 2 h following gavage, representing the proportion of cells capable of internalizing mCherry and forming large endosomal vacuoles. Error bars, means ± SEM; ****$P < 0.0001$; unpaired Student's $t$-test. The number of enterocytes = 376 cells from 14 wild-type and 341 cells from 14 Dr-Vsp$^{-/-}$ zebrafish. **i** Relative mCherry internalization profile comparing between wild-type and Dr-Vsp$^{-/-}$ zebrafish larvae at 2 h following gavage. Data were means ± SEM percentage of intracellular mCherry intensity along the specified axes, with a 100% indicates the maximal value of the average intensity on each axis. (Left) Apicobasal axis of individual cells. (Right) Longitudinal axis along the posterior portion of mid-intestine. Number of fish = 14 for each zebrafish line. Data were presented as mean ± SEM, collected from four independent experiments for both wild-type and Dr-Vsp$^{-/-}$.

early endocytic trafficking[46–49] and can function similarly to Vsp during active endocytosis. The presence of compensatory enzymes may account for the remaining endosomal vacuoles in Dr-Vsp$^{-/-}$ LREs (Fig. 6f, h); some Dr-Vsp$^{-/-}$ zebrafish larvae survived and thrived into adulthood despite seemingly suffering from nutrient malabsorption.

Another scenario considers Dr-Vsp's 3-phosphatase activity towards PI(3,4)P$_2$ and PI(3,4,5)P$_3$[9]. PI(3,4)P$_2$ has been found in recycling endosomes[50], where it may serve as a transient intermediate in the conversion of PI(3)P to PI(4)P during recycling endosome formation. Dr-Vsp at recycling endosomes could be involved in this process by dephosphorylating the intermediary PI(3,4)P$_2$[45]. Unless a compensatory mechanism is available, the loss of Dr-Vsp's 3-phosphatase activity can cause excessive PI(3,4)P$_2$ retention and, as a result, disrupt the formation of recycling endosomes during endocytic recycling in LREs. The defect may also reduce subsequent endocytosis when the internalized membrane and specific receptors are unable to fully return to the plasma membrane. Furthermore, based on a previous study showing that Vsp exhibits small but significant activity toward PI(3,5)P$_2$ by in vitro enzyme assay[51], Dr-Vsp may contribute to PI(3,5)P$_2$ homeostasis in the late endocytic pathway in LREs[5,45]. The precise mechanism by which Dr-Vsp is recruited to function at specific endomembranes has yet to be determined, and this will be one of the important issues to be addressed in the future for a better understanding of the molecular mechanisms underlying the physiological role of Vsp in LREs.

Two distinct endocytic pathways have been described in zebrafish LREs: (i) fluid-phase endocytosis for fDex uptake; and (ii) receptor-mediated endocytosis for mCherry uptake[23]. In our study, Dr-Vsp deficiency in zebrafish LREs affected fDex uptake more remarkably than mCherry uptake. Furthermore, following the gavage with fDex-mCherry mixture, the difference in the intracellular distribution of fDex and mCherry in LREs was more pronounced in Dr-Vsp$^{-/-}$ mutants (Supplementary Fig. 4). While the precise underlying mechanism remains unsolved and may require further investigations, our results seem to suggest that the contribution of Dr-Vsp to LRE endocytosis may differ between fluid-phase and receptor-mediated endocytosis, possibly

due to subtle differences in molecular mechanisms, such as PIP conversion, between the two pathways.

Our immunostaining results indicate that Dr-Vsp was also expressed in other tissues than the intestine. Among them, the expression in the testis and kidney is consistent with the results obtained from other species[11,12,14,16], indicating that Vsp is functionally conserved across animal species. We suspected that defective endocytosis in Dr-Vsp$^{-/-}$ LREs is most responsible for growth restriction and higher mortality in zebrafish larvae (Fig. 5), particularly during the critical period when zebrafish larvae rely primarily on exogenous feeding via intestinal absorption[28,31]. Although Dr-Vsp is also expressed in larval pronephros (Fig. 2a), Dr-Vsp deficiency in this organ is less likely to be the primary cause of mortality. This is because larval pronephros begins to function at around 48 hpf[52], and defective pronephros should exhibit phenotypes earlier than what was observed in our study. We have previously reported that Vsp regulates mouse sperm motility via its 5-phosphatase activity toward PI(4,5)P$_2$ on the plasma membrane during sperm capacitation[11]. Vsp deficiency in mice caused abnormal sperm movement and significantly reduced in vitro fertilization. Dr-Vsp is expressed in zebrafish testis (Fig. 1a)[7], but Dr-Vsp$^{-/-}$ zebrafish have no apparent fertility defect. Further research may be necessary to determine whether Dr-Vsp in zebrafish sperm serves a similar function as its ortholog in mouse sperm.

Vsp is a membrane protein that functions upon membrane depolarization[6–8]. Our study raises an open question of how Dr-Vsp is activated in LREs. According to the in vitro experiment, the voltage required to activate Dr-Vsp is beyond the physiological range of membrane potential in native cells[18]. Unfortunately, there is no current in vivo method for measuring the membrane potential of intracellular vesicles in zebrafish LREs. However, several possible mechanisms may induce Dr-Vsp activation in endosomes. First, unlike in vitro conditions, native Dr-Vsp in LREs may function at physiological membrane potentials, possibly through interactions with unknown endogenous auxiliary subunits, and a partial activation could be sufficient to induce Dr-Vsp phosphatase activity during active endocytosis. This is similar to what may happen in mouse sperm, where

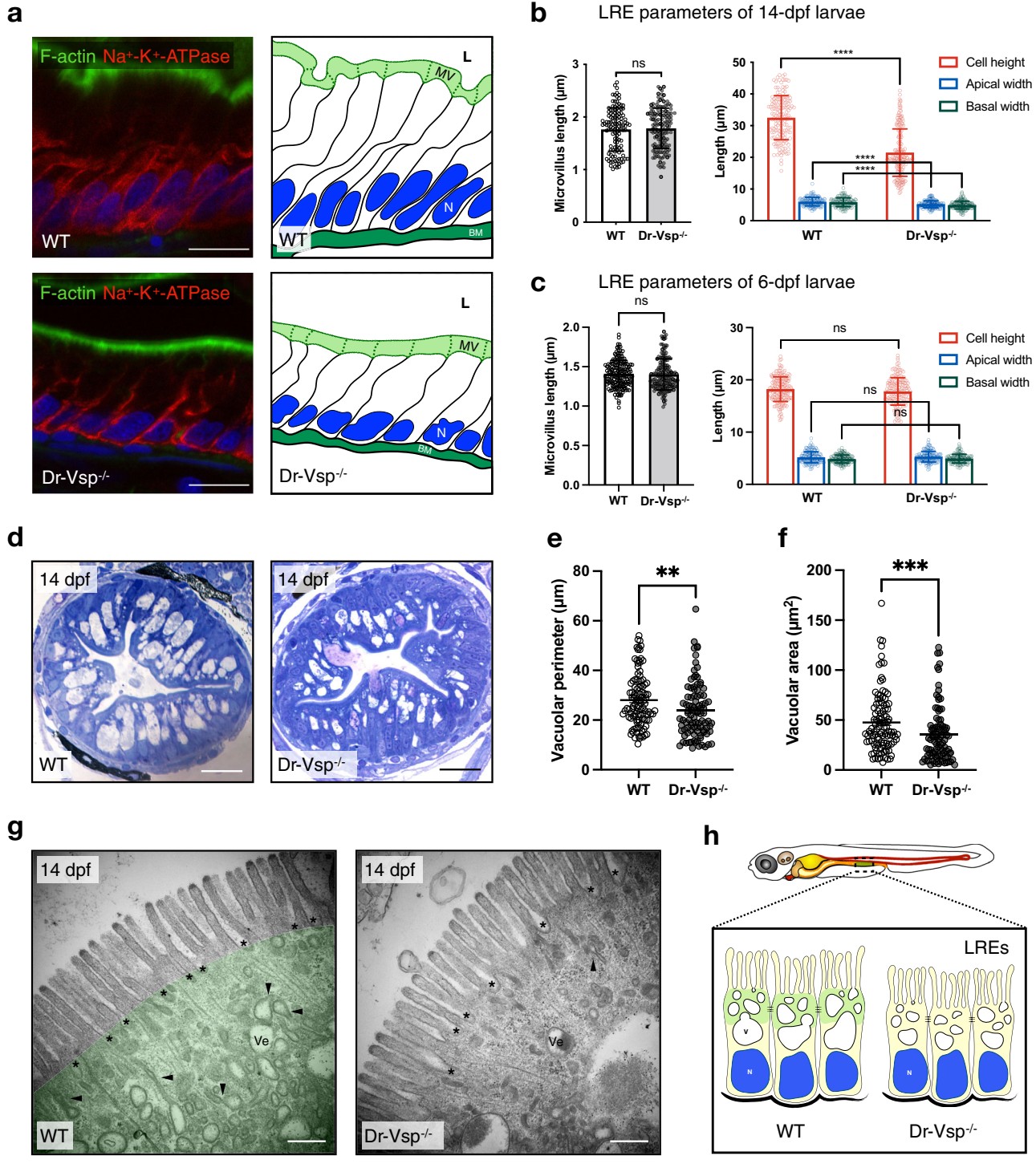

physiological membrane potential appears to activate mouse Vsp[11]. Second, membrane dissociation during endosomal vesicle formation could alter membrane potential probably by changing membrane capacitance, ultimately accelerating depolarization. Third, changes in micro-environments such as membrane tension, luminal pH, and opening of endomembrane ion channels[53–55] may facilitate Dr-Vsp activation under physiological conditions. Further in vivo analyses are required to explore the relationship between membrane potential, Vsp activation, and endosomal membrane trafficking. Potential future tests include rescue experiments of Dr-Vsp$^{-/-}$ mutants by expressing modified versions of Vsp, such as a voltage-insensitive or a voltage-

hypersensitive version, to study how membrane potential affects endocytosis in LREs. Additionally, optogenetics could be applied in zebrafish to manipulate membrane potential by specifically targeting photoactivated ion transporting molecules to specific membrane vesicles.

Finally, our comparative study reveals that Vsp is conserved in the digestive tracts of zebrafish and *C. intestinalis* Type A, as are other endocytosis-associated genes (Figs. 2e, 8d–f)[23,56,57], suggesting that Dr-Vsp and Ci-Vsp share a conserved endocytic function in the absorptive epithelial cells. Endocytosis-dependent nutrient absorption is conserved across species from unicellular organisms to vertebrates. Unicellular organisms acquire nutrients

**Fig. 7 Morphological change in Dr-Vsp$^{-/-}$ LREs. a** Immunostaining showing cellular framework in sagittal sections of 14-dpf wild-type (WT) and Dr-Vsp$^{-/-}$ LREs; and their corresponding schematic illustration. Green, F-actin. Red, Na$^+$-K$^+$ ATPase. Blue, DAPI. Scale bar = 10 μm. **b** Cellular parameters of LREs at 14 dpf comparing wild-type and Dr-Vsp$^{-/-}$ zebrafish larvae. (Left) Microvillus length. (Right) Cell height, apical width, and basal width. Enterocytes, ≥120 cells from five larvae for each zebrafish line. **c** Cellular parameters of LREs at 6 dpf comparing wild-type and Dr-Vsp$^{-/-}$ zebrafish larvae. (Left) Microvillus length. (Right) Cell height, apical width, and basal width. Enterocytes, ≥170 cells from six larvae for each zebrafish line. Error bars, means ± SD; ****$P < 0.0001$; unpaired Student's $t$-test; ns no statistically significant difference. **d** Representative transverse sections at mid-intestine of 14-dpf wild-type and Dr-Vsp$^{-/-}$ zebrafish larvae, showing LREs with intracellular supranuclear vacuoles. Sections were stained with toluidine blue. Scale bar = 20 μm. **e, f** Vacuole parameters of 14-dpf LREs. **e** Vacuolar perimeter of wild-type and Dr-Vsp$^{-/-}$ LREs. **$P = 0.0024$; Mann–Whitney $U$-test. Vacuoles, 108 for wild-type and 95 for Dr-Vsp$^{-/-}$. Data were collected from three larvae for each zebrafish line. **f** Vacuolar area of wild-type and Dr-Vsp$^{-/-}$ LREs from the same samples as in Fig. 6e. ***$P = 0.0003$; Mann–Whitney $U$-test. **g** Representative TEM images at mid-intestine of 14-dpf wild-type and Dr-Vsp$^{-/-}$ zebrafish LREs, showing the ultrastructures of microvilli and subapical region. Key features of absorptive enterocytes are presented, including membrane invaginations at inter-microvillous spaces (*), cytoplasmic tubules and tubule-vacuole complexes (arrowhead), and numerous endocytic vesicles (Ve). In wild-type LRE, green represents the region upper to the supranuclear vacuole, which corresponds to the area with a positive immunofluorescence signal of Dr-Vsp in Fig. 3a, b. Scale bar = 500 nm. **h** Schematic diagram of wild-type and Dr-Vsp$^{-/-}$ LREs. Dr-Vsp (green) is primarily localized in the cytoplasm close to the apical surface of enterocytes, promoting endosomal maturation during endocytosis-dependent nutrient absorption. N nucleus, V vacuoles.

from their surroundings by endocytosis, and many invertebrates show LRE-like gastrodermal cells that line the digestive tract to absorb nutrients by endocytosis[58]. Endocytotic absorption through epithelial cells has also been reported in the nutrient absorptive organs, such as the trophotaenia of viviparous fish embryos[59] and the ileum of pre-weaning mammals[60,61]. Furthermore, LREs have also been proposed to be involved in the trans-cellular transport of antibodies and other antigens across enterocytes to extra-intestinal tissues for immune modulation[23,31,61–63]. Because both LRE-like cells and Vsp gene are widely observed from Cnidaria to mammals[23,58,60,61,64], the endocytic role of Vsp may well be conserved among a wide range of animal species.

## Methods

**Zebrafish maintenance**. We used RIKEN wild-type zebrafish (*Danio rerio*) obtained from RIKEN Brain Science Institute (Saitama, Japan), and *Tg(tpte:EGFP)* transgenic zebrafish that express EGFP under the endogenous expression of *tpte*, the Dr-Vsp-encoding gene. The fish were fed a non-calorie-restricted standard diet twice daily and raised in an aquatic system maintained at 28 °C with a 14/10 h light/dark cycle.

**Biological materials for Ciona intestinalis's experiments**. Matured and young adult *Ciona intestinalis* type A (*C. robusta*) were collected during the spawning season at Port Chiba in Tokyo Bay (Chiba, Japan). *Ciona* juveniles were developed by artificial insemination using matured eggs and sperm obtained from dissected gonoduct. The specimens of whole-mount juveniles and dissected post-pharyngeal region of the digestive tracts of the young adult were prepared for in situ hybridization[39].

**Ethics**. Protocols used for the animal experiments in this study were approved by the Animal Research Committee of Osaka University, Japan (No. 27-079). All procedures were conducted in accordance with the regulation of the Animal Care Facility, Center for Medical and Translational Research (CoMIT) at Osaka University, Japan.

**RNA isolation and reverse transcription PCR (RT-PCR)**. Total RNA was purified from zebrafish larvae and adult tissues using TRIzol$^{TM}$ LS reagent (Invitrogen, USA) with volume adjustment for small samples. Total RNA (200 ng) extracted from each sample was reverse-transcribed using SuperScript$^{TM}$ III First-Strand Synthesis System (Invitrogen, USA). For semi-quantitative RT-PCR, fragments of *tpte* and *β-actin* were amplified from cDNA using the primer sets listed in Supplementary Table 1.

**Whole-mount in situ hybridization (WISH) for zebrafish larvae**. Zebrafish larvae were fixed in 4% PFA/PBS at 4 °C overnight. Next, the larvae were washed twice with PBS for 5 min each time, followed by briefly washed with 100% MeOH and kept in MeOH at −20 °C until the following steps. WISH was performed using a digoxigenin (DIG)-labeled RNA probe according to the standard protocol. In this study, the Dr-Vsp cDNA was subcloned into pBluescript II KS (+) plasmid vector. The resultant plasmid was linearized by the XhoI enzyme and later used as a template for synthesizing a DIG-labeled RNA probe using T3 RNA polymerase. At the beginning of the WISH process, fixed zebrafish larvae were treated with

proteinase K (10 μg/mL) for 15 min at room temperature. Prehybridization was done at 55 °C for 1 h. Larva samples were hybridized with 10 ng of DIG-labeled Dr-Vsp cRNA probe overnight. During post hybridization, larva samples were washed with PBS containing 0.2% Tween-20 and then incubated with anti-DIG-alkaline phosphatase (AP), Fab fragments (Roche, Mannheim, Germany) at 4 °C overnight. The samples were subsequently washed with PBS containing 0.2% Tween-20 every 20 min for 3 h to remove excessive antibodies. Positive signals of the hybridized Dr-Vsp mRNA transcript were visualized by color development using BCIP/NBT solution (Wako, Japan) for 2 h at room temperature. The stained larvae were observed under Zeiss Stemi SV11 Stereo Microscope (Zeiss, Berlin, Germany).

**In situ hybridization, detection, and observation of gene expression in Ciona intestinalis**. DIG-labeled antisense RNA probes for *Ciona* genes were synthesized from T7 RNA polymerase promoter-attached amplified cDNA, and probes were purified by centrifugal ultrafilter[13]. WISH of the *Ciona* specimens was performed using "InSitu Chip"[65]. Gene expression was visualized using BCIP/NBT solution (Roche, Mannheim, Germany). After color development, the digestive tracts of the *Ciona* juveniles were sectioned transversely with a thickness of 10 μm[39]. Whole-mount and sectioned specimens of *Ciona* were observed under a BX51 microscope (Olympus, Japan) or SZX12 Stereo Microscope (Olympus, Japan).

**CRISPR-Cas9-mediated transgenesis and mutagenesis**. *Tg(tpte:EGFP)* zebrafish line was generated via CRISPR-Cas9-mediated knock-in transgenesis[27] with adaptation to the *tpte* gene (Supplementary Fig. 1). The sgRNA for genome digestion contained a target site (AAAACCGCTACGTCTGCAGCAGG) derived from the *tpte* exon sequence located between the transcriptional start site and translational start site. Donor DNA for knock-in contained Mbait-hsp70 promoter-EGFP-polyA sequences. Successful knock-in transgenesis was validated by sequence mapping for donor DNA insertion and comparing EGFP signal with *tpte* expression pattern. Dr-Vsp-deficient (Dr-Vsp$^{-/-}$) zebrafish were generated via CRISPR-Cas9-mediated mutagenesis. Potential CRISPR targets for the *tpte* gene were identified via the online tool CHOPCHOP[66] based on genomic DNA sequence obtained from the Ensembl zebrafish genome database version 9 (Zv9, https://asia.ensembl.org/Danio_rerio/). We selected three targets close to the translation initiation site without potential off-target (Supplementary Table 2). In our setting, target T1 in exon 6 (Supplementary Fig. 3) yielded the highest efficiency in producing adequate frameshift mutation. Microinjection for Dr-Vsp mutagenesis was performed based on the reported protocol[67]. All injected embryos were incubated at 28.5 °C to support early development. Subsequently, some injected embryos underwent genome extraction and sequence analysis, while the remaining embryos were raised as the founder (F0) generation.

**Genomic DNA extraction and sequence analysis**. Genomic DNA was isolated from whole larvae or fin-clips of adult zebrafish using the HotSHOT DNA extraction procedure[68]. Genomic DNA fragments covering CRISPR targets were amplified by PCR using locus-specific primer sets listed in Supplementary Table 2. We used BigDye$^{TM}$ Terminator v3.1 Cycle Sequencing Kit (Thermo Fisher Scientific, Waltham, MA) for analyzing the sequences of PCR fragments.

**Fish sectioning, immunohistochemistry (IHC), and toluidine staining**. Zebrafish larvae were fixed in 4% PFA/PBS, followed by 30% sucrose/PBS, and embedded in 5% ultra-low gelling temperature agarose (Sigma-Aldrich, USA). Embedded specimens were cryosectioned in 10 μm slices using Leica CM3050S cryostat (Leica, Germany). Antibodies used for IHC are as follows: Anti-zebrafish Vsp/Tpte (1:500) (NeuroMab, USA), Acti-stain$^{TM}$ 488 Phalloidin (1:500) (Cytoskeleton, USA), Alexa Fluor$^{TM}$ 594 Phalloidin (1:500) (Invitrogen, USA), and anti-Na$^+$-K$^+$

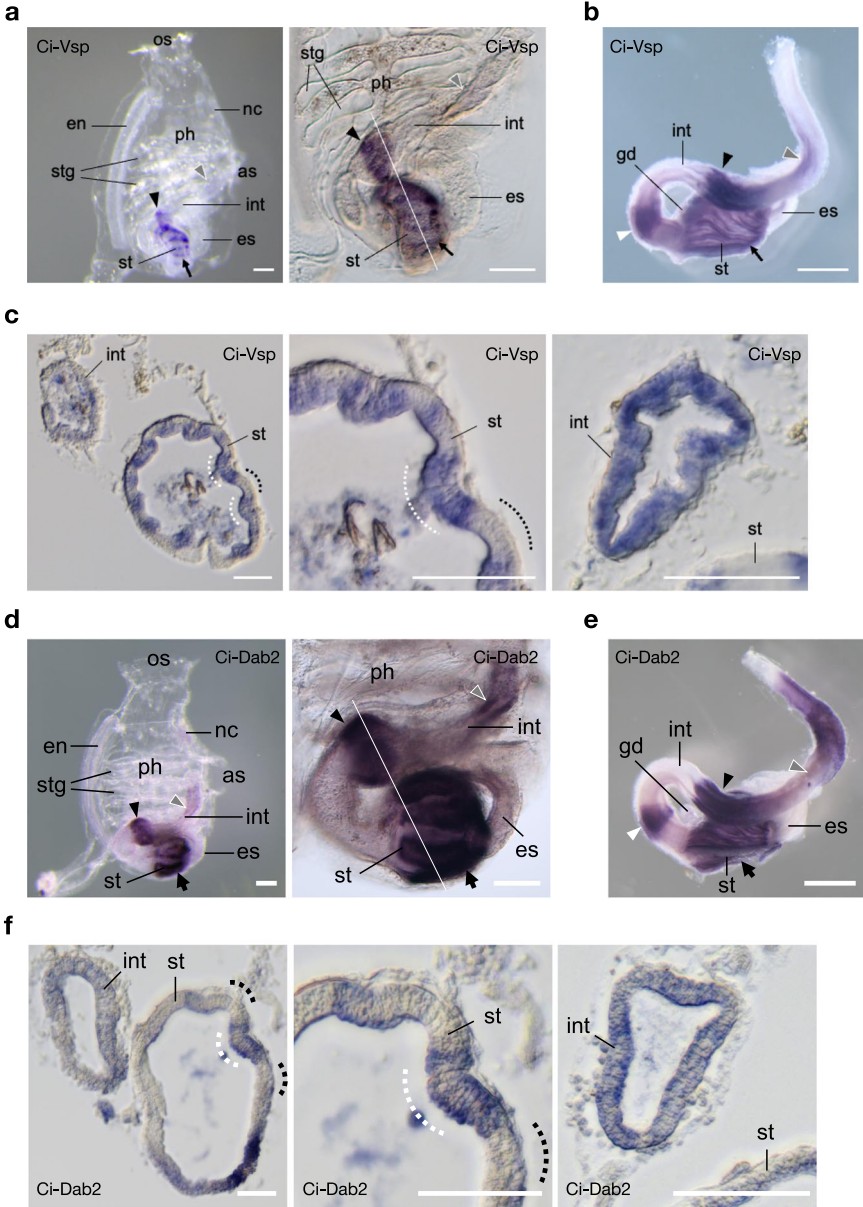

**Fig. 8 Expression profile of Ci-Vsp and Ci-Dab2 in ascidian *C. intestinalis*. a–c** Expression profile of Ci-Vsp by WISH. **a** Whole-mount (left) 2-week-old juvenile showing regional expressions in the stomach (st; arrow), bulged-region of the mid-intestine (int; arrowhead), and posterior-intestine (gray arrowhead). Higher magnification (right) reveals a stripe-like distribution of signals in the stomach and bulged intestine. **b** Dissected post-pharyngeal region of the digestive tract of the 1.5-month-old young adult showing an additional expression region (white arrowhead) in the looped-intestine.
**c** Transverse section specimens of the juvenile (white line in a) showing Ci-Vsp expressions in the epithelium of the digestive tract (left). Expression signals in the stomach epithelium are restricted mainly in the inner-folds (white dotted lines), which contain several absorptive enterocytes[38], as opposed to the outer-folds (dotted lines) with many pancreatic-related exocrine gene expressions[39]. Circumferences of the stomach (middle) and bulged intestine (right) show alternating epithelial morphology and Ci-Vsp expression. **d–f** Expression profile Ci-Dab2. **d** Whole-mount (left) 2-week-old juvenile showing regional expressions in the stomach (st; arrow), bulged-region of the mid-intestine (int; arrowhead), and posterior-intestine (gray arrowhead). Higher magnification (right) reveals a stripe-like distribution of signals in the stomach and bulged intestine. **e** Dissected post-pharyngeal region of the digestive tract of the 1.5-month-old young adult showing an additional expression region (white arrowhead) in the looped-intestine. **f** Transverse section specimens of the juvenile (white line in **d**) showing Ci-Dab2 expressions in the epithelium of the digestive tract (left). Expression signals in the stomach epithelium are restricted mainly in the inner-folds (white dotted lines) as opposed to the outer-folds (dotted lines). Circumferences of the stomach (middle) and bulged intestine (right) show alternating epithelial morphology and Ci-Dab2 expression. Scale bars = 200 μm in **a**, **d**, 1 mm in **b**, **e**, and 50 μm in **c**, **f**. OS oral siphon, AS atrial siphon, Ph pharynx, En endostyle, Stg stigma, Es esophagus, Nc neural complex, Gd gonad.

ATPase (1:500) (Ab76020; Abcam, USA). Images were taken under Zeiss LSM880 confocal microscope with Airyscan (Zeiss, Berlin, Germany). For toluidine blue staining, larvae were fixed in 4% PFA/PBS at 4 °C overnight, followed by 2.5% GA at 4 °C for 2 h and 1% GA overnight. Samples were subsequently dehydrated by ethanol substitution and embedded in Epon for thin sectioning before the staining process. The sections stained with toluidine blue were observed under a BX63 microscope (Olympus, Japan).

**Transmission electron microscopy (TEM)**. Zebrafish were fixed in 4% PFA at 4 °C overnight before en-bloc sectioning of the mid-intestine region. Sectioned samples were subsequently fixed in 2.5% GA at 4 °C for 2 h, followed by 1% GA overnight. Specimen preparation and staining were done as standard protocol, with technical support from the Center for Medical Research and Education, Osaka University. Briefly, the fixed samples were incubated with 1% osmium tetroxide, then dehydrated in ethanol series and propylene oxide. The samples were then

embedded in Epon and ultra-thin sectioned, followed by contrast stained with uranyl acetate and lead citrate. TEM images were acquired under the Hitachi H-7650 transmission electron microscope (Tokyo, Japan) operating at 80 kV and analyzed using ImageJ software (NIH, USA).

**Pre-embedding immuno-electron microscopy (Immuno-EM)**. Zebrafish larvae were fixed in 4% PFA/HEPES or 4% PFA/0.1% GA/HEPES before being embedded in 5% ultra-low gelling temperature agarose (Sigma-Aldrich, USA) and cryosectioned in 10 μm slices using Leica CM3050S cryostat (Leica, Germany). Sectioned samples were incubated in 5% normal goat serum (NGS)/0.125% saponin/PBS for 30 min, then incubated in anti-zebrafish Vsp/Tpte antibody (1:200) (clone N432/21, NeuroMab, USA) at 37 °C for 2 h, followed by Alexa Fluor® 488 FluoroNanogold antibody (1:30) (Nanoprobes, USA) at room temperature for 2 h. Immunogold particles were intensified at room temperature with HQ SILVER™ enhancement kit (Nanoprobes, USA) and 5% selenium toner. Samples were further treated with 1% osmium tetroxide, 0.5% uranyl acetate, and underwent the same process as described in the TEM protocol.

**Cell culture and transfection**. MDCK type 2 (MDCKII) cells (CRL-2936, ATCC) were cultured in D-MEM high glucose (Wako, Japan) containing 10% FBS at 37 °C under 5% $CO_2$. Transient expression was achieved by a co-transfecting mCherry-Dr-Vsp plasmid with (i) EGFP-Rab5 plasmid (early endosome marker) or (ii) EGFP-Rab11 plasmid (recycling endosome marker) into MDCKII cells using Lipofectamine™ 3000 transfection reagent (Invitrogen, USA).

**Fish gavage and live imaging**. 6-dpf zebrafish larvae were gavaged with Alexa Fluor 488-tagged dextran 10,000 MW (fDex) solution (Molecular Probes, USA) or purified mCherry solution (prepared in the laboratory) following the established method[32] (see also Fig. 6a). A microforged capillary needle loaded with either solution was directly inserted through the larval mouth to inject the solution into the anterior intestinal bulb. Gavaged larvae were incubated at 28.5 °C for 2 h to allow intestinal absorption before live image acquisition under Zeiss 710 confocal microscope (Zeiss, Berlin, Germany) with temperature control at 28.5 °C for entire sessions. The following parameters were analyzed: (i) the percentage of cells with fDex- or mCherry-filled vacuoles per section; and (ii) relative fDex or mCherry internalization profile measured across the apicobasal axis of individual cells and the longitudinal axis along the posterior portion of mid-intestine. The data of (ii) were analyzed by measuring fluorescence intensity using Plot Profile function in ImageJ software (NIH, USA)[23] and presented as means ± SEM percentage of intracellular fluorescence intensity along the specified axes, with a 100% indicates the maximal value of the average intensity on each axis.

**Statistics and reproducibility**. Sample sizes, definitions of replicates, and statistical details for each experiment are reported in the figure legends. Data are presented as mean ± SEM unless otherwise specified. $P$ value < 0.05 was regarded as statistically significant. Graphs and statistical analyses were performed using Prism 8 (Graphpad Software, San Diego, CA).

**Parameter analysis of larval enterocytes**. Quantification of cellular dimensions of LREs was done based on the previous procedure[33]. We performed immunostaining with Acti-stain™ 488 Phalloidin (1:500) (Cytoskeleton, USA) and anti-Na$^+$-K$^+$ ATPase Antibody (1:500) (Ab76020; Abcam, USA) to outline individual cell (Fig. 7a). In the mid-intestine region, each confocal image was selected where enterocytes appeared the broadest, and the following parameters were analyzed: central height, apical width, basal width, and microvillus length.

**Body length and survival analysis**. Wild-type and Dr-Vsp$^{-/-}$ zebrafish were raised in the standard system of the Animal Care Facility, Center for Medical and Translational Research (CoMIT), Osaka University. Body length and remaining number of larvae were measured periodically from the day of fertilization to 28 dpf. All data were collected from three sets of independent experiments.

**Analysis of gene expression level using RNA-seq data**. Raw data of the RNA-seq experiments were retrieved from the GEO Datasets database with accession number GSE124970[23]. Heatmap representing expression levels of selected genes in IECs and LREs (Fig. 2e) was analyzed using the web-based application iDEP v0.92 (http://bioinformatics.sdstate.edu/idep92)[69].

**Image processing**. Raw images were processed and analyzed using ImageJ software (NIH, USA).

**Reporting summary**. Further information on research design is available in the Nature Research Reporting Summary linked to this article.

## Data availability
Data generated or analyzed in the present study are included in this published article, as well as its Supplementary Information and Supplementary Data 1 files. Key reagents and resources are listed in Supplementary Table 3. Further information and requests should be directed to and will be fulfilled by the corresponding author.

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

## Acknowledgements

We thank Dr. Shinji Kanda (University of Tokyo, Japan) for helpful advice on this study. Our thanks are also due to Drs. Junji Takeda, Akihiro Harada, Shin-ichiro Yoshimura, Atsushi Tamura, Hiroo Tanaka, Yoshifumi Okochi, Akira Kawanabe (Osaka University, Japan), and all laboratory members for valuable suggestions and discussions. We thank Mr. Eiji Oiki (Osaka University, Japan) for advice in electron microscopy; Mr. Masayuki Yamagishi and Mr. Satoshi Nakayama (Chiba University, Japan) for advice and technical assistance in the in situ hybridization of the *Ciona* specimens. Technical support for this study was provided by the Center for Medical Research and Education and the Center for Medical and Translational Research (CoMIT) at Osaka University. This work was supported by Grants-in-Aid from JSPS (12J01957 and 15K18575) (to T.K.), Grants-in-Aid from Ministry of Education, Culture, Sports, Science, and Technology (MEXT) and JSPS (15H05901 and 25253016 to Y.O.), JSPS (16H02617 to T.K. and Y.O.), Core Research for Evolutional Science and Technology, Japan Science and Technology Agency (CREST, JST) (JPMJCR14M3 to Y.O.), and the Mitsubishi Foundation (to Y.O). M.M. was supported by Grans-in-Aid from MEXT (A00H059010 and A15H059010).

## Author contributions

Conceptualization and methodology, A.R., T.K., and Y.O.; Investigation, A.R., M.M., S.H., Y.K., F.T., T.M., M.I.H., and M.O.; Formal analysis, A.R., T.K., and Y.O.; Visualization and writing—original draft, A.R.; Writing—review and editing, A.R., T.K., Y.O.; Funding acquisition, T.K., Y.O., M.M., A.R.; Supervision, Y.O. All authors have read and agreed to the final version of the manuscript.

## Competing interests

The authors declare no competing interests.

**Additional information**

