## [Peer Review File · Communications Biology]

Reviewers' comments:

Reviewer #1 (Remarks to the Author):

In this excellent study from Ratanayotha et al., the authors demonstrate that voltage-sensing phosphatase (VSP) play an evolutionarily conserved role in absorptive enterocytes by regulating endosome dynamics and nutrient absorption. While VSPs had previously been shown to function in sperm, their physiological functions in other tissues has remained less studied. Moreover, biophysical characterization of VSPs had largely been conducted in heterologous expression systems—as such the physiological function of endogenous VSPs in particular subcellular compartments has remained a topic of debate with some focusing on VSP function in the plasma membrane while others note that VSPs are rather enriched intracellularly in endosomes. The authors use a diverse set of state-of-the-art approaches to elucidate the physiological roles of VSPs in enterocytes. Their findings will be of broad interest to those interested in digestive physiology, endosome dynamics, and VSPs.

To show specific enrichment of VSP expression in lysosome-rich enterocytes (LREs), the authors first use rt-PCR from various tissues and show gut expression, followed by in situ and generation of a transgenic zebrafish inserting EGFP into the 5' untranslated region of the endogenous VSP gene to show specific enrichment in LREs. Immunohistochemistry using a previously developed VSP antibody and immunogold-TEM shows subcellular enrichment of the VSP protein below the plasma membrane and around the intracellular vacuole. This localization is further explored in MDCK cells by demonstrating colocalization with early (rab 5) and recycling (rab11) endosomes.

To assess the functional role of VSPs in LREs, the authors generated a loss-of-function mutant by introducing a frame-shift mutation in an early exon using CRISPER/Cas9 and showing loss of protein expression using immunohistochemistry. VSP^{-/-} mutants show growth and survival phenotypes underscoring the importance of this protein for organismal physiology. By gavaging mCherry and dextran, the authors demonstrate endocytic deficits in VSP^{-/-} mutants and TEM shows a much less complex endocytic tubular network in mutants. Finally authors show that the enrichment of VSPs in absorptive enterocytes is conserved, showing similar enrichment in urochordate intestines.

Feedback for improving the manuscript:

- Figure 4 Green annotation on the lower set of photomicrographs should be changed to RAB11. VSP-mCherry images are a little hard to see. Perhaps change to another colour rather than red, and annotate image with colocalized vesicles and those that do not colocalize. Presumably there are recycling endosomes in the rab5 image that have mCherry but not rab5 and the converse in the rab11 image.
- In Suppl Fig 2 Please comment on the immunogold dots that appear in enlargement 3—is this basal lamina?
- In Suppl Fig 3 For VSP mutant, please indicate how many amino acids prior to the stop codon either in this figure legend or in the methods section.
- In Figure 5, it would be preferable to show individual datapoints because the description in the text suggests that some of the larvae had significant growth retardation while others did not—this would be better represented with individual datapoints for body length.
- Figure 6 f and h Y-axis label is confusing because it suggests that some cells are affected in VSP^{-/-} mutants while others are not but photomicrographs seem to show that all cells in the epithelium are similarly affected.
- Please show individual datapoints on bar graphs—similar to how figure f and h are shown.

Reviewer #2 (Remarks to the Author):

In manuscript by Ratanayotha et al. the authors investigated the function of the voltage sensitive phosphatase (VSP) in zebrafish. They characterized the expression and localization of tpte (a.k.a. vsp) using in-situ hybridization, CRISPR mediated knock-in of a reporter gene (eGFP) into tpte promoter and using antibodies. They found that tpte is highly expressed in Lysosome Rich Enterocytes (LREs) and that the protein (Tpte) localizes to subapical membranes. Further, using a CRISPRs they generated a mutant for the gene (tpte^{-/-}) and assayed its function in LREs. The authors found that tpte is involved in the uptake of endocytic cargos in LREs as well as growth and survival of larval zebrafish. Gavage experiments performed using a mixture of cargoes and TEM analyses led them to conclude that tpte is involved in endosomal maturation in LREs. Additionally, analysis of LRE morphology suggest a role of tpte in maintenance of key morphological features of Lysosome-rich enterocytes including endosomal structures. Lastly, the authors show that tpte expression is conserved in ascidians and overlaps with enterocytes with endocytosis-dependent absorptive function.

This is an interesting manuscript that uncovers a role for the lipid phosphatase VSP (Tpte in zebrafish) that is relevant for the understanding of intestinal physiology. Their observations and the known activity of VSP open the way for future mechanistic studies on the cell biology of LREs. Further, their findings suggest that both VSP and LRE function are conserved across chordates. Overall, the experimental data is of good quality and generally supportive of their conclusions. However, there are a few relatively minor points that need to be addressed.

1. The authors used co-gavage of f-Dex and mCherry to assay LRE activity and claim that these cargoes are somehow differentially affected in tpte mutants. While this may be indeed the case, it may also result from kinetic issues and the properties of the specific fDex probe used. The authors should either better support their claim with more quantitative, time-course analyses or revise their conclusions.
2. Is the antibody signal detected by IF lost in tpte mutants?
3. Panel 2E was generated using data published in another study. While the reference is provided, the provenance of the primary data used for the panel is not clearly indicated in the text, e.g., "We analyzed transcriptome data from LREs published by..."
4. Nomenclature of the gene. The authors use Dr-VSP to refer to the zebrafish gene. However, the correct name that adheres to the nomenclature is tpte for the gene and Tpte for the protein. Please correct throughout the manuscript.
5. The colocalization of overexpressed protein in MDCK (Figure 4b) is only suggestive but does not decisively represent the endogenous scenario in LREs. It would be better to move the figure to supplementary and tone down the claim that Tpte is localized on Rab5+ endosomal membranes in LREs. Nevertheless, the localization to endosomal membranes is clear from the endocytic assays.
6. The body length measurements shown in the graph (Figure 5c) do not reflect what is shown in the adjacent representative images of 28dpf fish, is there a scale problem? Replacing with a better representative image would be best.
7. Figure 2d and supplementary Figure 3e: It seems either the green (Tpte) channel is missing or not labelled.
8. To improve the readability, it would be useful to label the various domains in supplementary figure 3d.
9. Line 68: The word 'protein' seems to be missing.

Response to the reviewers' comments

Manuscript ID: COMMSBIO-22-1438

“Voltage-sensing phosphatase (VSP) regulates endocytosis-dependent nutrient absorption in chordate enterocytes”

We would like to thank the reviewers for carefully reading and giving us insightful comments to improve the quality of our manuscript. The corresponding changes have been made accordingly, and our point-by-point response to the reviewers' comments are listed as follows.

Reviewer 1

1. Figure 4 Green annotation on the lower set of photomicrographs should be changed to RAB11. VSP-mCherry images are a little hard to see. Perhaps change to another colour rather than red, and annotate image with colocalized vesicles and those that do not colocalize. Presumably there are recycling endosomes in the rab5 image that have mCherry but not rab5 and the converse in the rab11 image.

Response: Thank you for pointing out this mistake. The green annotation on the lower panel of Figure 4b has been changed to Rab11. We have changed the color representing Dr-VSP from red to magenta to improve visibility, and annotated images with (1) circles for colocalized vesicles; and (2) arrowheads for non-colocalized Dr-VSP signals. As the reviewer suggested, recycling endosomes may be present in Rab5 image, and early endosomes may be present in Rab11 image, presumably presented by non-colocalized Dr-VSP signals in both images. We have included this explanation in the legend of Figure 4b, which can be found on **page 21; lines 677-683**.

2. In Suppl Fig 2 Please comment on the immunogold dots that appear in enlargement 3—is this basal lamina?

Response: Enlargement 3 in Supplementary Figure 2 depicts the basal part of LREs and the basal lamina lining the outer surface of the larval intestine. As the reviewer noted, some immunogold-like particles are present in this area. The particles in this area were scattered equally with low density both inside and outside the tissue, thus leading us to believe that they are likely non-specific signals. We have included this explanation in the figure legend, which can be found on **page 25; lines 813-815**.

3. In Suppl Fig 3 For VSP mutant, please indicate how many amino acids prior to the stop codon either in this figure legend or in the methods section.

Response: Dr-VSP^{-/-} mutant protein contains 120 amino acids. The N-terminus (before CRISPR target) contains 73 amino acids that are unchanged from the wild-type Dr-VSP, whereas the subsequent 43 amino acids are altered protein with early termination, missing the

voltage-sensing residues (in S4) and the cytoplasmic catalytic region. In the revised manuscript, we have created a new figure of the Dr-VSP^{-/-} mutant model, simulating how the mutant protein would appear in topological view, as shown in Supplementary Figure 3e. The description has also been provided in the figure legend on **page 25; lines 832-837**.

4. In Figure 5, it would be preferable to show individual datapoints because the description in the text suggests that some of the larvae had significant growth retardation while others did not—this would be better represented with individual datapoints for body length.

Response: Individual data points for survival rate and body length of zebrafish larvae have been included in Figures 5a and 5b, respectively.

5. Figure 6 f and h Y-axis label is confusing because it suggests that some cells are affected in VSP^{-/-} mutants while others are not but photomicrographs seem to show that all cells in the epithelium are similarly affected.

Response: We appreciate the reviewer for pointing this out. We are sorry for some confusing presentation in Figure 6. We first note that Y-axis in figures 6f and 6h represents the percentage of enterocytes capable of internalizing fDex (or mCherry) and forming endosomal vacuoles. These graphs provide a bird's-eye view of the effect of Dr-VSP deficiency on the entire LRE segments, whereas the photographs in Figures 6d and 6e focus on representative segments. Our findings suggest that, while all Dr-VSP^{-/-} mutant LREs lose VSP activity and are seemingly defective in endocytosis, a small proportion of cells can still internalize the gavaged fDex and mCherry to some degree, possibly due to the presence of compensatory mechanisms, as we proposed in the Discussion. To improve clarity, we have included an additional description in the Result on **page 6; lines 176-178**. We have also revised the Y-axis labels in figures 6f and 6h, as well as their legends on **page 22; lines 715-717 and 728-729**, respectively. The revised figure legends are shown below:

Figure 6 (f) Percentage of the enterocytes containing fDex-filled vacuoles at 2 hours following gavage, representing the proportion of cells capable of internalizing fDex and forming subsequent endosomal vacuoles.

Figure 6 (h) Percentage of the enterocytes containing mCherry-filled vacuoles at 2 hours following gavage, representing the proportion of cells capable of internalizing mCherry and forming large endosomal vacuoles.

6. Please show individual datapoints on bar graphs—similar to how figure f and h are shown.

Response: Individual data points have been shown on bar graphs in Figures 7b and 7c, as well as in Supplementary Figures 5d and 5e.

Reviewer 2

1. *The authors used co-gavage of f-Dex and mCherry to assay LRE activity and claim that these cargoes are somehow differentially affected in tpte mutants. While this may be indeed the case, it may also result from kinetic issues and the properties of the specific fDex probe used. The authors should either better support their claim with more quantitative, time-course analyses or revise their conclusions.*

Response: We thank the reviewer for pointing this out. The reviewer raised concerns about the LRE activity assay that used co-gavage of fDex and mCherry probes. We agree that other factors such as kinetics or properties of the probes could also affect the results, and determining the precise mechanism would require additional analyses. Indeed, we intend to conduct a new series of experiments focusing on this topic in our next project, but the preparation is still in its initial stages. In response to the reviewer's comment, we have toned down the statement throughout the manuscript and revised our descriptions in the Result on **page 6; lines 195-196**, and in the Discussion on **page 11; lines 341-344**, which are now as follows:

Results: The findings also *raise the possibility* that Dr-VSP potentially plays a greater role in fluid-phase endocytosis, the primary mechanism of fDex uptake into cells.

Discussion: While the precise underlying mechanism remains unsolved and may require further investigations, our results *seem to suggest* that the contribution of Dr-VSP to LRE endocytosis may differ between fluid-phase and receptor-mediated endocytosis, possibly due to subtle differences in molecular mechanisms, such as PIP conversion, between the two pathways.

We acknowledge the reviewer's suggestion and hope to be able to clarify this issue after further experiments.

2. *Is the antibody signal detected by IF lost in tpte mutants?*

Response: In the immunofluorescence, the antibody signal for Dr-VSP/TPTE was not detected in Dr-VSP^{-/-} mutant LREs, as shown in Supplementary Figure 3e of the original manuscript. However, the annotation for the green channel (Dr-VSP) was missing from the original figure, so we have added the annotation to the revised version. Please also note that a new Supplementary Figure 3e of the Dr-VSP^{-/-} mutant model has been added, and the original Supplementary Figure 3e has been renamed 3f in the revised manuscript.

3. *Panel 2E was generated using data published in another study. While the reference is provided, the provenance of the primary data used for the panel is not clearly indicated in the text, e.g., “We analyzed transcriptome data from LREs published by...”*

Response: Following the reviewer's advice, we have revised our description in the Results on **page 4; lines 124 – 126**, and the figure legend on **page 20; line 652**, to clarify the provenance of the primary data used to generate Figure 2e.

4. *Nomenclature of the gene. The authors use Dr-VSP to refer to the zebrafish gene. However, the correct name that adheres to the nomenclature is tpte for the gene and Tpte for the protein. Please correct throughout the manuscript.*

Response: The reviewer raised concerns about the gene and protein nomenclature. We have changed the gene name from *vsp* to *tpte* in our revised manuscript to comply with the reviewer's suggestion. As for the protein name, however, we would like to keep the name of VSP because it properly reflects its molecular function and has been commonly used in physiological studies, with over 30 related publications (from 2005 to 2022) listed on PubMed.gov. Humans have two paralog genes: *TPTE* (also called *PTEN2*) and *TPIP* (also called *TPTE2*). In most species, *vsp/tpte* gene is unique and is more homologous to *TPIP* than to *TPTE*. In other words, *TPIP* gene is the conserved gene among species, whereas *TPTE* gene is only unique to the human genome (duplicated after the branchpoint between *Homo sapiens* and Chimpanzee). Results of phosphatase assay revealed that human TPTE does not show phosphoinositide phosphatase activities and, in fact, human TPTE protein has amino acid changes in the consensus Cx5R sequence at the enzyme active center. Human TPTE's molecular nature remains unclear. Although the gene name of *TPTE* was originally used for a historical reason because it was described as the first human protein with similarity to PTEN, the naming of *TPTE* is very misleading in terms of molecular functions. For further detailed discussion, please see our review (Okamura et al., 2018).

In our revised manuscript, we introduced the name Dr-VSP/TPTE for zebrafish VSP in the Introduction on **page 3; line 85**, to clarify the meaning. We hope this explanation clarifies the reviewer's concern.

5. *The colocalization of overexpressed protein in MDCK (Figure 4b) is only suggestive but does not decisively represent the endogenous scenario in LREs. It would be better to move the figure to supplementary and tone down the claim that Tpte is localized on Rab5+ endosomal membranes in LREs. Nevertheless, the localization to endosomal membranes is clear from the endocytic assays.*

Response: The reviewer suggested that we move the result of the MDCK model (Figure 4b) to the Supplementary section because of its indecisive representation of LREs.

We agree that the result is still only suggestive. However, we would like to keep the result as one of the main figures because we believe it provides good evidence to support potential function of VSP at the endomembranes. VSP on plasma membranes of heterologous expression systems has been characterized for many years, and the concept of VSP function at the endomembranes is one of the main points in this study. To obtain more direct evidence, we are now planning to investigate the detailed localization of VSP in native LREs using specific antibodies and immunoelectron microscopy, as well as transgenic zebrafish with fluorescence-labeled endomembranes.

Following the reviewer's suggestion, we have toned down our interpretation regarding the results of the MDCK model in the Result on **page 5; line 142**, and in the Discussion on **page 9; line 284**, in our revised manuscript, as shown below:

Results: The Dr-VSP signal was broadly co-localized with Rab5-positive and Rab11-positive intracellular compartments (Figure 4b, Supplementary Movie 1, Supplementary Movie 2), *suggesting* that Dr-VSP is potentially expressed on the membranes of early and recycling endosomes.

Discussion: Results from the MDCK model *seem to suggest* that Dr-VSP is enriched at the early and recycling endosomal membranes (Figure 4b), consistent with our recent in vitro study in which fluorescent signal from mCherry-fused Dr-VSP expressed in HEK293T cells was also distributed throughout the intracellular compartments (Kawanabe et al., 2020).

We hope this explanation meets with the approval of the reviewer.

6. *The body length measurements shown in the graph (Figure 5c) do not reflect what is shown in the adjacent representative images of 28dpf fish, is there a scale problem? Replacing with a better representative image would be best.*

Response: Thank you for bringing this point to our attention. After rechecking, we realized that our original Figure 5c does not properly reflect the body length measurement shown in Figure 5b, so we have replaced with new representative images on a calibrated scale in the revised version of Figure 5c. We confirm that our original data were accurately measured, and we have also revised Figure 5b to show individual data points.

7. *Figure 2d and supplementary Figure 3e: It seems either the green (Tpte) channel is missing or not labelled.*

Response: We apologize for the confusion in this part. The green channel for Dr-VSP/TPTE was included in the original Figure 2d and Supplementary Figure 3e, but its annotations were missing. We have added the annotations to both revised figures. As in our response to the reviewer's comment 2, the original Supplementary Figure 3e has been renamed 3f in the revised manuscript.

8. *To improve the readability, it would be useful to label the various domains in supplementary figure 3d.*

Response: We have revised Supplementary Figure 3d and its legend to show the Dr-VSP topology model with various functional domains in the cytoplasmic region. The revised figure legend can be found on **page 25; lines 828-831**.

9. *Line 68: The word 'protein' seems to be missing.*

Response: The word 'protein' has been added accordingly on **page 2; line 68**.

Reference

- Kawanabe, A., Mizutani, N., Polat, O. K., Yonezawa, T., Kawai, T., Mori, M. X., & Okamura, Y. (2020, May 4). Engineering an enhanced voltage-sensing phosphatase. *J Gen Physiol*, 152(5), 1-11. <https://doi.org/10.1085/jgp.201912491>
- Okamura, Y., Kawanabe, A., & Kawai, T. (2018, Oct 1). Voltage-Sensing Phosphatases: Biophysics, Physiology, and Molecular Engineering. *Physiol Rev*, 98(4), 2097-2131. <https://doi.org/10.1152/physrev.00056.2017>

REVIEWERS' COMMENTS:

Reviewer #1 (Remarks to the Author):

The authors have produced a thoughtful and thorough revision of their manuscript in response to reviewer feedback.

Reviewer #2 (Remarks to the Author):

The authors have addressed all experimental issues reasonably. The only point that is not OK is the nomenclature, there is no gene or protein Dr-VSP/TPTE. It is ok to introduce the history of the gene name and any uncertainties of its phylogenetics. However, the convention is clear and should be:

Gene: lower case and italics, *tpte* (could add a.k.a. *vsp* if necessary).

Protein: Tpte or Vsp

Response to the reviewers' comments

Manuscript ID: COMMSBIO-22-1438A

“Voltage-sensing phosphatase (Vsp) regulates endocytosis-dependent nutrient absorption in chordate enterocytes”

We would like to thank the reviewers once more for their helpful comments on our revised manuscript. The corresponding changes have been made, and our point-by-point responses to the reviewers' comments are listed below.

Reviewer #1

The authors have produced a thoughtful and thorough revision of their manuscript in response to reviewer feedback.

Response: Thank you for your comment.

Reviewer #2

The authors have addressed all experimental issues reasonably. The only point that is not OK is the nomenclature, there is no gene or protein Dr-VSP/TPTE. It is ok to introduce the history of the gene name and any uncertainties of its phylogenetics. However, the convention is clear and should be:

Gene: lower case and italics, tpte (could add a.k.a. vsp if necessary).

Protein: Tpte or Vsp

Response: Thank you for your advice. The protein names VSP and TPTE have been changed to Vsp and Tpte, respectively, throughout the revised manuscript. To improve clarity, we have also introduced the protein name Tpte, along with Vsp, in the Introduction on **page 3; lines 69**, which is now as follows:

“Voltage-sensing phosphatase (Vsp, also known as Tpte) is among the key molecules that regulate PIPs' homeostasis. Encoded by *tpte* gene, Vsp is a unique membrane protein with two functional domains: ...”